



# Communicating public avalanche warnings – what works?

Rune V. Engeset[1,2], Gerit Pfuhl[1], Markus Landrø[1,2], Andrea Mannberg[1], Audun Hetland[1]

[1]UiT The Arctic University of Norway, Hansine Hansens veg 18, 9019 Tromsø, Norway.
[2]Norwegian Water Resources and Energy Directorate, Oslo, Box 5091 MAJ., 0301, Oslo, Norway.

*Correspondence to*: Rune V. Engeset (rue@nve.no)

**Abstract.** Like many other mountainous countries, Norway has experienced a rapid increase in both recreational winter activities, and fatalities, in avalanche terrain during the past few decades: during the decade 2008-2017, 64 recreational avalanche fatalities were recorded in Norway. This is a 106 % increase from that of the previous decade. In 2013, Norway therefore launched the National Avalanche Warning Service (NAWS), which provide avalanche warnings to transport and
preparedness authorities, and to the public. Previous studies suggest that avalanche warnings are used extensively in trip and preparedness planning, and have a relatively strong influence on the decisions people are taking in order to reduce risk. However, no evaluation, concerning how efficiently the warnings are communicated and understood, has been done to date. Avalanche warnings communicate complex natural phenomena, with a variable complexity and level of uncertainty about both the future and the present. In order to manage avalanche risk successfully, it is fundamental that the warning message can be
understood and translated into practice by a wide range of different user groups. Less avalanche-competent users may need simple information to decide when to stay away from avalanche terrain, while professional users may need advanced technical details in order to make their decisions. To evaluate how different modes of communication are understood, and how efficiently the informational content is communicated, we designed and implemented a web-based user survey. The modes of presentation were based on the Varsom.no 2017-version (Varsom.no being the national portal for natural hazards warnings in Norway).
We first asked a panel of experts from NAWS to answer the survey, and used their answers to establish the indented comprehension of the avalanche warning. We thereafter recruited over 200 recreational users, and compared their answers to those of the experts for the different modes of communication. Our empirical analyses suggest that most users find the warning service to be useful and well suited for their needs. However, the effectiveness of a warnings seems to be influenced by the competency of the user and the complexity of the scenarios. We discuss the findings and make recommendations on how to
improve communication of avalanche warnings.

## 1 Introduction

Does the Norwegian Avalanche Warning Service effectively communicate its intended message? Risk communicators should pursue their intention to assess whether the message they disseminate is appropriate, understandable and useful (Charrière and Bogaard, 2016). This is a matter of prime concern during a period of dramatic change in information technology and
information consumption in the society. The internet is rapidly becoming the main source of information, and studies show





that communication is important, e.g. Brigo et al. (2016) concludes that internet campaigns with emotional content is important to effectively promote awareness programs on risk of avalanches and increase public knowledge related to these persisting and serious threats. This study focus less on campaigns and more on the avalanche warnings and forecasts published daily by Avalanche Warnings Services (AWS').

## 1.1 Public avalanche warnings

In order to prevent avalanche accidents, AWS' throughout the world publish avalanche warnings to the preparedness authorities and the public. The standards for publishing danger levels and structuring the information in the warnings have been developed over the years with the aim to provide the users with a product that is as effective as possible. Most AWS' use the systems devised by the European AWS' (www.avalanches.org, EAWS, 2017, Műller et al., 2016) or the North American AWS' (www.avalanche.org, Statham et al., 2010, Statham et al., 2018). All AWS' quantify the danger into five levels (1-5), and use one or more of these standard elements: 1) a main (flash) message, 2) most avalanche prone terrain (elevation; aspects), 3) avalanche problems, 4) snow cover and avalanche history, and 5) avalanche danger assessment and prognosis. The products from the different AWS' varies considerable in degree of detail, use of text, symbols and graphics, degree of advice provided etc. (Burkeljca, 2013a). However, most avalanche warnings are typically structured in a standard journalistic inverted pyramid approach (Scanlan, 2000, Burkeljca, 2013b), where the most important information is presented at the top. More detailed and advanced information is sequentially presented further down in the pyramid. Accordingly, the standard EAWS approach (see www.avalanches.org) presents the danger level at the top, often accompanied by a flash message (a short main message). Secondly, the core zones (the most avalanche prone terrain) are pointed out, typically by describing which elevation intervals and compass directions (sectors) that have the highest danger. At this level, or at the level below, the current avalanche problems are described, followed by a description of the avalanche danger, snow cover and avalanche history, weather history and prognosis, and finally observations from the field. The pyramid approach also reflects what is useful to users at three different levels of competence (Mitterer et al., 2014): the top level of information mainly targets all users, especially beginners with limited ability to understand and use complex information and users who are after getting the key information quickly, the medium level targets users with an intermediate to advanced knowledge of avalanche and snow assessments, while the detailed bottom level of information in mainly useful to experts.

The danger level range from 1 – low to 5 – very high (extreme) and is an expression of the probability and size of expected avalanches in a given geographical region over a given period of time. In order to derive a danger level, the geographical extent should be above 100 km$^2$. It is a generalisation over a larger area, which typically have significant local variability (Jamieson et al., 2008, Schweizer et al., 2008, Techel et al., 2016). The European Avalanche Danger Scale (EADS; EAWS, 2016) was introduced in 1993 (SLF, 1993) and is used by all European AWS' but the Swedish AWS. The avalanche warning is a prognosis of expected danger over time, typically a period of 24 hours, and is based on an analysis of the current snow cover and the effects of the weather on the snow and avalanche conditions during the prognosis period. The avalanche problems (Atkins, 2004, Landrø et al., 2013, Statham et al., 2018) describe the characteristics of the avalanche danger in more detail: The type





of avalanche (dry or wet, slab or loose), trigger and failure mechanism, expected terrain locations, predictability and ease of detection. The level of details varies between AWS', as do the number of categories. Advice for back country travellers or preparedness authorities are provided by some AWS', either in the flash message (what to be aware of or do), as part of the avalanche problem (specific advice; is the problem manageable, and if so, how) or avalanche danger level (general advice).

The snow cover and avalanche analysis provides a description of the snow properties and distribution relevant to avalanche conditions (e.g. snow height, recent snow fall, surface, wet/dry, critical layers, etc.) as well as recently observed avalanches in the region (e.g. numbers, sizes, types, failure planes, etc.). The avalanche danger assessment provides further details on avalanche threats, the distribution within the region, effects of expected weather, uncertainties, etc.

Introducing the EADS in 1993 as a European standard (Meister, 1995) improved communication of avalanche danger, and
provided a basis for rule-based management strategies. The danger level is used by many users (Winkler and Techel, 2014; Steiermark, 2015; Procter et al., 2014) and affects decision making during backcountry tours (Techel et al., 2015, Furman et al., 2010) and risk management authorities. Avalanche warnings provide important information for backcountry tour planning as well as en-route (Winkler and Techel, 2014; LWD Steiermark, 2015; Baker and McGee, 2016).

## 1.2 Warning and risk communication

Impact-based hydro-meteorological hazard warnings are getting more and more in demand, as are impact-based warnings for other geohazards. Impact-based warnings do not only assess threat and danger, but also exposure and vulnerability (WMO, 2015), and have been shown to be more effective than other types of warnings (DeJoy, 1999) The purpose of warnings is to inform people at risk about the hazard and to promote "correct", and safe behavior (Wogalter et al., 2005). Informed decision-making leads to desirable outcomes and prevents unnecessary costs to society (Pielke and Carbone, 2002). Avalanche warnings
provide users with both general and specific information about the current and expected level of avalanche danger, the type of avalanche problem at hand, and with behavioral advice. The main aim of the warning message is to inform the user about the nature and severity of current and expected threats, and about how he or she can mitigate the risk or avoid the threats. However, since most regional AWS' do not provide specific and local descriptions of the forecasted risk, it may be difficult to effectively reach this goal. In addition, most AWS' lack detailed information on the type and number of individuals who are at risk, and
on the exposure and vulnerability of these. Thus, most AWS' provide impact-based warnings in a general sense, but not in terms of impact specific to detailed geographical locations, people, roads and so on.

Although risk communication research has been a growing field since the 1980s (Sivle, 2016), some researchers report that warning practices have not changed much during the past decades (Kasperson, 2014), and there is sometimes a gap between the intended message (warning) and the message received (Gigerenzer et al., 2007). A range of factors contributes to this gap.
One such factor is that many people find it difficult to interpret numbers and probabilities. People's ability to make meaning out of numbers and statistics is often referred to as numeracy (Lipkus and Peters, 2009). Both large-scale surveys and small-scale experiments show that many individuals lack this ability (Låg et al., 2014; Kirsch al., 2002), and that even well-educated individuals often display a low level of numeracy (Lipkus et al., 2001). One approach to mitigate this problem is to use



qualitative explanations with words like "likely", and "unlikely". Even though people vary in their understanding of such words, users can conceptualize the concepts by comparing them to risks they already understand (Gordon-Lubitz, 2003, Edwards and Elwyn 2001).

Another reason for a mismatch between the intended and received message is that people vary in their motivation to use, and

competence to read warnings. The level of use and understanding of the information provided in the warnings vary between different user groups, and between different geographical regions (Wogalter et al., 1997). Geographical differences are driven both by differences in the characteristics of the user groups present in the area, and by differences in the complexity and amount of supporting information provided by the regional AWS's (Burkeljca, 2013a). Differences in search and use of warnings may further be driven by variations in the level of trust in authorities and experts, and by personal experiences of natural hazards

(Wachinger et al. 2013). Avalanche danger may in fact be so complex that a novice will not be able to manage the same terrain as experts, no matter how well the warning is communicated. The avalanche warnings are communicating low probability phenomena, which may in itself reduce the engagement at the user' side and the interest in reading and using the avalanche warning, and reduce the interest in investing in understanding the warning. Another challenge is that the warnings are used in several different ways, which could lower the interest.

Taken together, these differences make it difficult for providers of avalanche warnings to meet the needs of all groups. The challenge facing providers of avalanche forecasts is made even more difficult by the lack of research on how efficient different ways of presenting the avalanche danger to different groups are. For example, less competent and motivated users may need simplified explanations and direct travel advice in order to be able to use the information. They may easily be overloaded if the forecast contains a lot of detailed information (Maltz, 2000; Liang et al., 2006). For advanced users, on the other hand,

simplified information and advice may be of limited use. Instead this group may demand detailed information about the snow cover. It can be challenging to simultaneously satisfy the needs of both groups.

**1.3 The Norwegian Avalanche Warnings Service, Varsom.no and RegObs**

During the past few decades, Norway, as many other countries, has experienced a rapid increase in recreational winter activities in avalanche terrain (mainly ski touring, snowmobiling and to some extent snowshoeing). The increase in backcountry

recreation has unfortunately been associated with an increase in fatal avalanche accidents. During the decade 2008-2017, avalanches claimed 64 recreational fatalities (61 % occurred in North-Norway and Svalbard.), the corresponding number for the decade 1998-2007 was 31 (NGI, 2018). By contrast, avalanche fatalities in houses and during transportation have decreased from 7 in 1998-2007 to 2 in 2008-2017 (NGI, 2018). Similar trends are reported from other parts of the world (e.g. Techel et al., 2016). Another 3 fatalities, all recreational, were recorded during 2018 winter season.

In order to halter the undesirable trend in avalanche accidents, the Norwegian Avalanche Warnings Service (NAWS) was established in January 2013 (Engeset, 2013). NAWS publishes regional avalanche warnings for Norway, including Svalbard, on a daily basis on the web portal www.varsom.no (Johnsen, 2013). The Norwegian Water Resources and Energy Directorate



owns and operates NAWS, in collaboration with the Norwegian Public Roads Authorities and the Norwegian Meteorological Institute.

In 2017, regional avalanche warnings were issued for 21 regions in Norway (Fig. 1). In addition, warnings were issued for the rest of the country when the avalanche danger was expected to reach danger level 4 or 5. An example of an avalanche warning

5 on Varsom.no is shown in Fig. 2. The avalanche warning published on Varsom.no includes the elements described in Fig. 2 and Table 1.

**Figure 1.** An example of the avalanche warning regions for Norway (screen dump from Varsom.no).



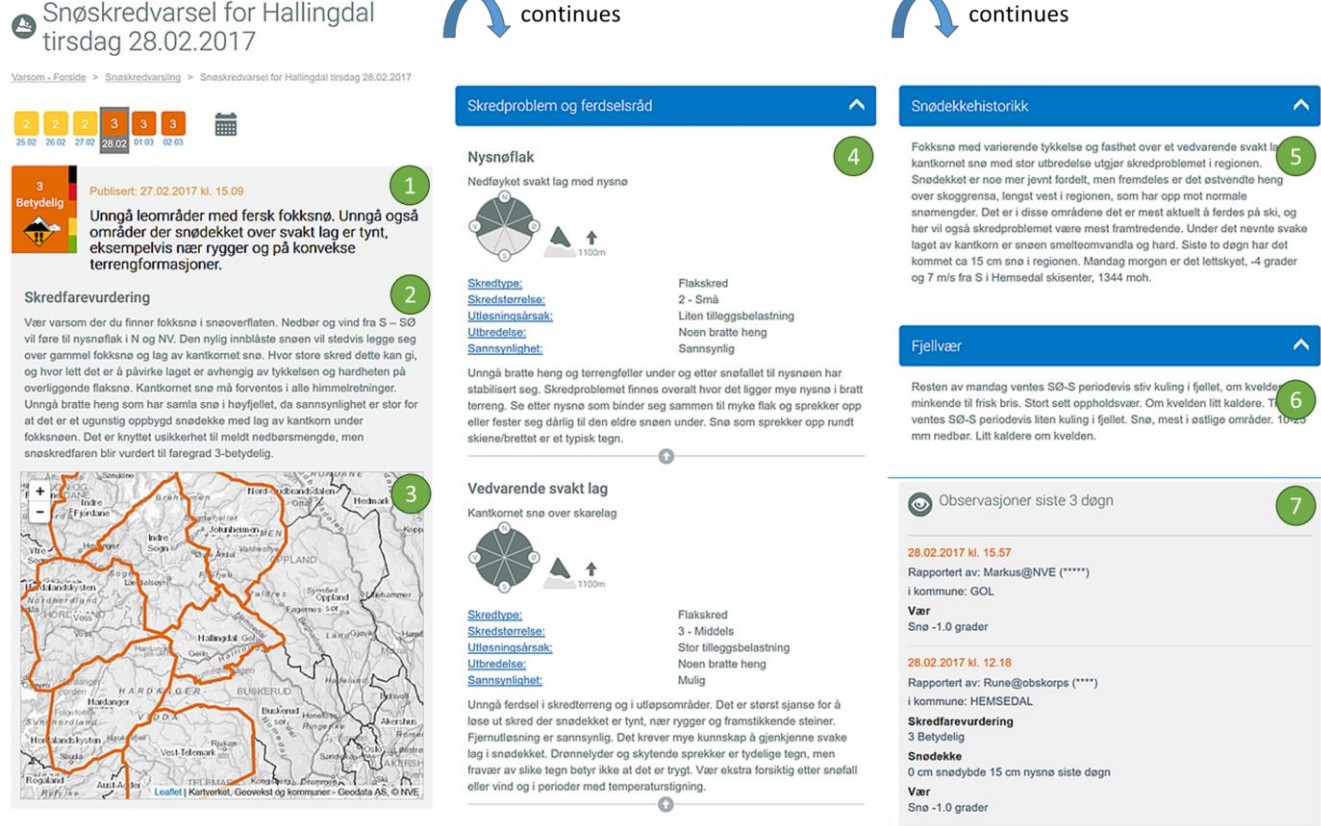

**Figure 2.** An example of an avalanche warning as issued on Varsom.no in 2017. The numbers refer to the elements analysed in this study. (1) Danger level and main message, (2) avalanche danger assessment, (3) region map, (4) avalanche problems, (5) snow cover history, (6) mountain weather prognosis and (7) RegObs-feed with observations. The figure shows the screen dump from a smartphone, with the middle and right panels showing the screen as the user scrolls down the page.

Four elements (danger level and main message, region map, avalanche problems, and mountain weather prognosis) are also available in an English version of the warning on Varsom.no, while two elements are in Norwegian only (avalanche danger assessment and snow cover history) and one is partly in English (RegObs-feed with observations).





**Table 1.** A description of the elements included in the avalanche warning on Varsom.no. Fig. 2 shows how the elements are shown in relation to each other on a smart phone web browser.

| Element | Description |
| --- | --- |
| 1 | The danger level, as described below as a combination of the probability and size of expected avalanches in the region during the forecast period. The flash, the main message displayed next to the danger level, and is a short statement of what constitutes the hazard and what the advice to the user is. This text is supposed to be very short and to the point, as if the user did not care to or have the competence to read the rest of the warning. |
| 2 | The avalanche danger assessment, which is a more detailed description of the avalanche hazard and what is the reason for it. It often includes a more detailed description of the uncertainty and local variability. |
| 3 | A map of the region, showing its extent and perimeter. |
| 4 | The avalanche problems, which at the time were storm slab, loose dry, wind slab, wet slab, wet loose, persistent slab, and glide avalanches (Landrø et al., 2013). A number of properties are forecasted for each avalanche problem: expected (destructive) size (1 to 5), expected additional load (natural, low or high), distribution (isolated, few, some or many steep slopes), release probability (possible, probable and likely) and core zone. Each avalanche problem has a predefined travel advice according to danger level. A main characteristic of the avalanche problem is that the properties of the weak layer are specified for slab-type avalanche problems, according to the Systematic snow cover diagnosis system (Kronthaler et al., 2013). The different avalanche problems have danger-specific advices for the users: How predictable and easy to detect is the problem in the field? Where in the terrain is it easy to trigger or be caught by avalanches from this problem? How to reduce the vulnerability to the problem? What should preparedness stakeholders be aware of? |
| 5 | The snow cover and recent avalanche history. This is a mixture of observation and an analysis of the snow cover at the initial time of the forecast period. It is an important baseline for making a prognosis of how forecasted weather may affect the avalanche danger during the forecasting period. |
| 6 | The mountain weather. This is the weather prognosis accessed at the time of writing the warning, and is thus the basis for prescribing the avalanche danger in combination with the snow cover history. |
| 7 | A real-time feed of observations submitted to and shared by the RegObs system. Regobs is the national system for sharing field observations in real-time (Ekker et al., 2013). |

Since the start in 2013, NAWS has continuously worked to improve both the competence level of observers and forecasters, and the system for presenting the forecast. User feedback suggested that most users find the warnings useful and of high quality. However, to date, no formal evaluation has been done of how effective NAWS is at communicating its intended message. Such an evaluation is important, because since public avalanche warnings have only been available in Norway since



2013, Norwegian users are likely less competent at using the warnings to manage risk than users in countries with a longer history of public avalanche warnings.

In order to improve the avalanche skills and knowledge in the Norwegian population in general, and the ski touring and snowmobiling communities in particular, NVE launched the "Snøskredskolen" (the avalanche school) on Varsom.no. The
avalanche school is a tailor-made resource for users of the avalanche warnings, as all key terms and concepts are explained and safe travelling advice is provided. It is also a much used resource for avalanche course providers.

As a system in the Varsom.no portfolio, RegObs provides data from the field as a basis for making forecasting decisions. RegObs is an open web- and app-based system for reporting, storing, querying and sharing observations and assessments from the field with the forecasters and the public. The observations are public and a live-feed of observations is displayed on
Varsom.no, next to the regional forecast. As such, RegObs is an integral part of Varsom.no and the communication of avalanche warnings. RegObs communicates the field observations and assessments, which the warnings are based on in a transparent way. As far as we are aware of, RegObs is the only open-access online real-time distribution system for avalanche forecasting right now, although previous efforts have provided open access to accident data (Duclos et al., 2008).

## 1.4 Aims of the study

The current study is a part of a larger project, which focuses on communication of flood, landslide, and avalanche danger warnings. In this study, we evaluate the efficiency of warnings by the NAWS on the website Varsom.no. Avalanche warning systems are used in trip and preparedness planning, and have been shown to have an influence on the decisions people are taking in order to reduce risk (e.g., Furman et al., 2010; Marengo et al., 2017). Mountain guides, course providers, rescue services and avalanche observers report that people actively respond to the avalanche warnings on Varsom.no, and to a large
degree choose snow, terrain and day for travelling according to the danger level, avalanche problem and advice provided by the NAWS.

Warnings should therefore ideally be revealing and unambiguous. To assess whether the warnings published by NAWS fulfil these requirements, we asked the following research questions: 1) Which risk factors are considered as most difficult to assess and manage? 2) Which elements in the warning are considered as most and least important? 3) Which elements are easily
misunderstood or considered poorly communicated? 4) What kind of information and features are missing or ignored by users? To answer these questions, we tested if users interpreted the danger and behavioural implications differently depending on if the message was described by text, by symbols or by pictures. Furthermore, we tested how well the warnings were understood, by testing four alternative ways of communicating two different danger scenarios.

## 2 Methods and data collection

We developed a web-based questionnaire and survey with test to collect data for the study. Questionnaires are useful tools for acquiring information on public knowledge and perception of natural hazards, and can provide valuable information to



emergency management agencies for developing risk management procedures (Bird, 2009). This chapter presents the methods, participants and survey design of this study.

## 2.1 Participants

### 2.1.1 Expert survey

The data in this study comes from two web-based surveys conducted during the autumn of 2017. The aim of the first survey was to derive a set of "correct" answers to questions on the meaning of the presented warnings. We therefore invited 200 avalanche experts (mainly avalanche forecasters and observers in the NAWS) to participate in the survey during the period 15-26 October, 2017. 110 experts provided complete responses. Of these, 67 were observers, 21 forecasters and 22 were not active in the forecasting. The last group consisted of former forecasters and observers, and of individuals with a professional

liaison role in the forecasting services. About 25 % of the expert respondents were female.

### 2.1.2 User survey

The purpose of the second survey was to test how well the NAWS message was understood by non-expert users, and therefore targeted recreational users, and potential users, of the NAWS. We recruited participants via social media, varsom.no and different user related web pages.

A total of 485 respondents answered the user survey. Not all respondents answered questions in all sections, leaving 361 respondents for analysis of Section B (avalanche warning), 222 respondents for analysis of Section C (text versus symbols and pictures) and 177 respondents for analysis of Section D (comprehension). The lower number of user respondents completing sections C and D reflects a common challenge in web-based surveys to engage participants enough to answer complex and time-consuming questions.

Of those proving details about gender, 17 % of the participants were women and 83 % were men. Mean age in the sample was 35 years (min = 19, max = 69). 26 % of the respondents lived in North-Norway, 8 % in Trøndelag, 11 % at the Northwest coast, 24 % at the West coast, 27 % in the South-East, 1 % on Svalbard, and 2 % answered other.

## 2.2 Survey design

In order to obtain valid responses and avoid careless responding, it is important that participants are motivated to take the
survey, understand all questions, feel that they can answer the questions, and do not lose interest before the end of the survey (Meade and Craig, 2012). We therefore pretested and revised all survey items in an iterative process. In the first stage, NAWS personnel, both forecasters and observers, provided qualitative feedback on how well the avalanche warnings communicated the message that NAWS' would like to disseminate, and this was taken into account when the questions and response alternatives were designed. We thereafter asked a test panel consisting of project members (N=12) to provide iterative feedback
on the content and structure of the survey. Based on the feedback from the NAWS personnel and the test panel, we rephrased



several questions and instructions to improve clarity. We also reduced both the number of questions and response alternatives. The latter shortened the completion time of the survey to about ten minutes. The survey was constructed so that it was possible to view and answer all questions using a variety of devices, including smartphones. The general structure and purpose of each section of the expert and recreational user surveys are described below.

5  Five sections (A-E) were identical in the expert survey and the user survey. Sections B, C and D provided the core data for the analysis in this paper. An overview of the survey is provided in Table 2.

**Table 2.** Overview of the survey.

| Section | Purpose |
|---|---|
| A | Collect background information related to avalanche training and competence, e.g., association with NAWS, if any, level of avalanche competence and training, activity level in terms of travelling in avalanche terrain, and level of comprehension of avalanche terrain. |
| B | Understand how respondents evaluate the available elements in the warning and how these are communicated (which elements are most important, least important, difficult to assess and manage, poorly communicated or easily misunderstood, and which elements are missing). |
| C | Test how users perceive three different ways of presenting the avalanche danger: text, symbols, and pictures. |
| D | Test the comprehension of two scenarios:<br>• Danger level 2 (wind slab problem), based on the warning for Troms region on 18 April 2017, and<br>• Danger level 4 (wet slab problem), based on the warning for Troms region on 4 April 2017.<br>For each scenario, the participant was first randomly presented with one out of four alternative ways of communicating the danger:<br>1. Avalanche danger level with explanation (general advice associated with the danger level),<br>2. Avalanche problem with technical details and advice (advice on how to manage the problem),<br>3. Avalanche problem with technical details only, and<br>4. Avalanche problem with advice only.<br>We thereafter asked the respondent to interpret and evaluate the warning in terms of 1) behavioural implications (based on a pre-defined set of options), 2) how well the avalanche warning was presented, and 3) how the respondent would describe the warning to others, and what travel advice s/he would give to them. |
| E | Collect background information related to demographics, and backcountry recreation, e.g., gender, age, home region, terrain activities, and use of avalanche gear and forecast. |

10  The two scenarios in Section D were based on the warnings (in Norwegian) on Varsom.no at the time of the survey (autumn 2017). The four alternatives given for each of the two scenarios in Section D where picked randomly for each user respondent.



We did not counterbalance the order, all respondents received first the level 2 scenario than level 4, but with different alternatives.

## 2.3 Expert survey

As briefly mentioned above, the main purpose of the expert survey was to derive a template of "correct" answers. More
specifically, we wanted to identify key information elements, and define sets of behavioural implications at different avalanche danger scenarios. In other words, we used the experts to operationalize the intended content of the avalanche forecasts. To make sure that the operationalization was valid, we used a relatively large and heterogeneous group of avalanche experts.

To limit completion time and mental strain for participants, we only used two avalanche danger scenarios (level 2 and level 4, see section D in Table 1). Each expert was randomly exposed to one out of four alternative ways to present the forecast for
each danger level: 1) only the avalanche danger level and very limited travel advice, 2) an avalanche rose, probability, distribution, expected size and type of avalanches, along with more detailed travel advice, 3) only an avalanche rose, probability, distribution, expected size and type of avalanches, or 4) only detailed travel advice (see figure 3, below).

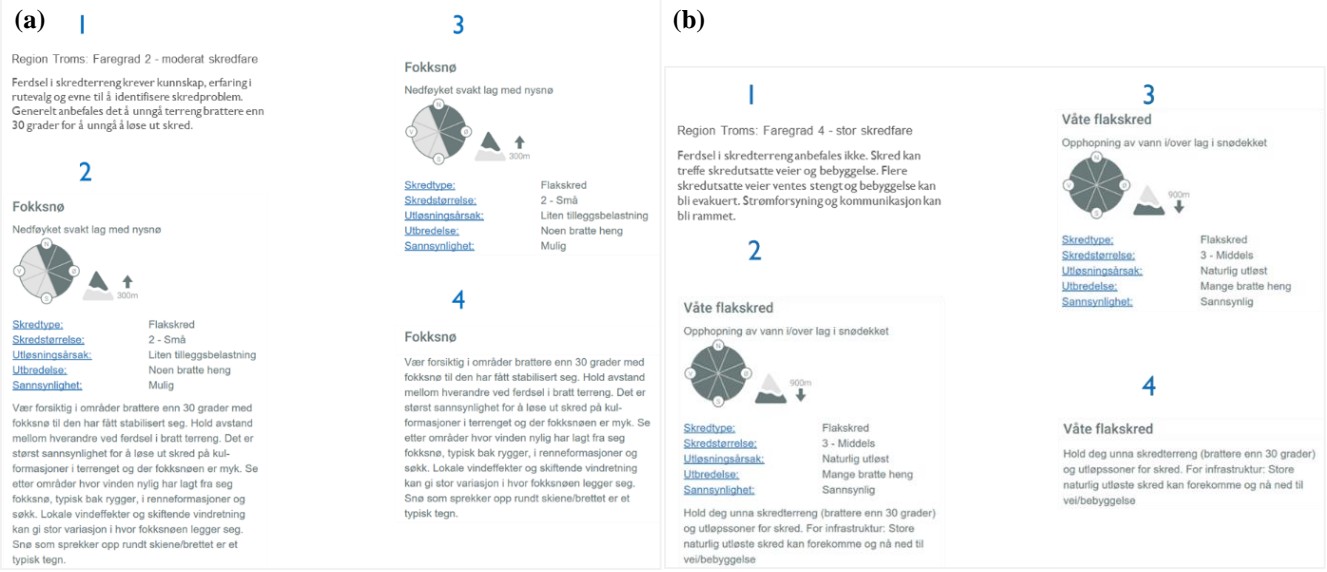

**Figure 3.** Alternatives 1 to 4 used for the two scenarios: (a) level 2 and wind slab (left) and (b) level 4 and wet slabs.

After the expert respondent had read the example, we first asked him or her to rate how well the danger was communicated in the example, on a scale from 0 to 10. We thereafter asked the expert to identify key information elements and behavioural implications of the avalanche forecast. The options were predefined, as described in Table 3. We were specifically interested in identifying the most important message that the forecast aimed to communicate.
In order to establish a communication effectiveness score, we used the expert answers to allocate weights to the different behavioural implications. We allocated positive weight of +1 to elements positively identified as important by more than one





out of three experts (33 %), and a weight of -1 to elements positively identified by less than one out of 5 experts (20 %). All other elements were given a weight of null. The expert choices and resulting weights are listed in Table 3. As can be seen in table 3, many experts agreed on the most important implications, and very few items are therefore close to the cut-off value. Nevertheless, to ensure that our results do not hinge on our chosen levels (33% and 20%), we have tested both upward and downward variations of the cut-off values. The results presented in section 3 are robust to these variations.

**Table 3.** Expert survey results and design weights established for a communication effectiveness score.

| Statement | 2-scenario response | 2-scenario weight | 4-scenario response | 4-scenario weight |
|---|---|---|---|---|
| 1. Unngå alle løsneområder (*Avoid all release areas*) | 20 % | -1 | 84 % | +1 |
| 2. Unngå noen løsneområder (*Avoid some release areas*) | 63 % | +1 | 9 % | -1 |
| 3. Unngå alle utløpsområder (*Avoid all runout areas*) | 8 % | -1 | 84 % | +1 |
| 4. Unngå noen utløpsområder (*Avoid some runout areas*) | 39 % | +1 | 11 % | -1 |
| 5. Unngå skredutsatte veier (Avoid avalanche-exposed roads) | 6 % | -1 | 75 % | +1 |
| 6. Kunne mye om snø for å vite hva jeg skal unngå (*Know a lot about snow in order to know what to avoid*) | 29 % | 0 | 16 % | -1 |
| 7. Grave i snøen for å vite hva jeg skal unngå (*Dig in the snow in order to know what to avoid*) | 12 % | -1 | 6 % | -1 |
| 8. Vite mye om været siste to dager for å velge terreng (*Know a lot about the weather the last two days in order to choose terrain*) | 45 % | +1 | 13 % | -1 |
| 9. Forvente store lokale forskjeller (*Expect large local variability*) | 71 % | +1 | 16 % | -1 |

## 2.4 User survey

The user survey was open to the public during the period 1 November – 15 December. We published links to the survey on a relatively wide set of platforms: Varsom.no, the free online skiing magazine friflyt.no, and on the Facebook page of the most popular weather service in Norway, YR.no. The association of snow scooter clubs (Skuterklubbenes fellesråd) and the Norwegian Hiking Association (DNT) kindly distributed the survey to their members. Finally, we announced the survey on the Nordic avalanche conference in Åndalsnes in the beginning of November.

Each participant was asked to answer the full survey (section A-E). In section D, the recreational users were, just like the experts, randomly exposed to one out of four alternative ways of presenting the avalanche forecast for the level 2 and level 4 scenarios, and thereafter to first rank how well the danger was communicated on a scale from 1 to 10, and to mark the most important behavioural implications of the forecast.



We used the weights in table 3 to calculate a "communication effectiveness score" for each participant, and each behavioural implication. To illustrate, consider a user who ticked the boxes for statements 1, 2 and 3 after reading an example of the 2-scenario. Based on the scores in table 3, we would give this user a score of -1. If the user instead ticked the boxes for statements 2, 3 and 4 after reading an example of the 2-scenario, we would give him or her a score of +1. The scores for the 2-scenario ranged from -4 to +4, and for the 4-scenario from -6 to +3.

## 2.5 Ethics

This study registered anonymous information exclusively and did not collect data that can be used to identify individuals. All respondents actively gave their consent for the use of the data for research and the project.

## 3. Results

In this chapter, we present the avalanche-related demographics of the user respondents (Section A and E), well-functioning and malfunctioning parts of the 2017-version of the avalanche warnings on Varsom.no, as perceived by the participants (Section B), the participants' evaluation of how well text, symbols and pictures assist the informational content in the warnings (data from Section C), the participants' evaluations of how well different levels of complexity in the text persuade the informational content in the warnings (data from Section D), and test results for level of comprehension at different levels of complexity in the warning texts (also data from Section D).

## 3.1 Demographics

The statistics of the users with respect to competence, experience, activities and geography are listed below and in Table 4:

- 14 % of the recreational respondents had no or little avalanche knowledge, 27 % stated that they had some avalanche related competence but no formal training, 48 % stated that they had avalanche related competence and formal training, and 10 % were avalanche instructors or professionals.
- 82 % stated that they had used avalanche gear (e.g., avalanche beacon, shovel and probe) for several seasons, 7 % one season only, and 11 % had never used this type of equipment.
- The majority of participants stated that their main activity in avalanche terrain was alpine ski touring (66 %). Relatively many respondents also stated that they engaged in off-piste skiing (32 %), or Nordic mountain skiing (23 %), while relatively few said that they travel in avalanche terrain by foot (9 %), on a snowmobile (7 %), or on snow shoes (3 %). Three percent stated that they engage in other types of activities in avalanche terrain. Note that the respondents could chose multiple activities.
- Concerning the use of NAWS, 76 % of the recreational users answered that they always use the avalanche warnings, 21 % use the warnings on a regular basis, and 3 % answered that they rarely read the forecast.



**Table 4.** Contingency table of respondents' experience (number of ski tours per year) versus competence.

| Competence | Experience (ski tours per year) | | | |
| --- | --- | --- | --- | --- |
| | **0** | **< 5** | **5-15** | **> 15** |
| None | 10% | 48% | 35% | 8% |
| Competent, no course | 6% | 23% | 41% | 30% |
| Competent, course | 0% | 6% | 39% | 55% |
| Expert | 0% | 5% | 14% | 82% |

## 3.2 Avalanche warning

A total of 361 user respondents completed the questions in Section B. In this section, we asked the respondents to identify risk
5   factors that they perceived difficult to manage or mitigate, parts of the avalanche warnings that they perceived difficult to
understand, and important information perceived to be missing in the avalanche warnings.

### 3.2.1 Avalanche risk factors considered difficult to assess and manage

In order to find out what the users consider as being most difficult to assess and manage, we asked "Which factors are most
difficult to assess and manage in order to complete a safe trip?". The respondents could choose multiple factors. Available
10   factors and results are shown in Fig. 4. The results show that

- The vast majority of both recreational participants and experts (87 % of the experts and 79 % of the recreational
  users) perceive that the *snow cover* is the single most difficult factor to assess and manage. This judgement does not
  depend on the respondent's experience or competence (Chi-square test, p = .516 and p = .403, respectively),
- More than every second expert (52 %) perceive that *other people in the group* is the most problematic factor, while
15   32 % of the recreational users rated this as the most difficult.
- Among recreational users, there is a relatively even distribution of individuals who perceive that *terrain traps* (26
  %), and *weather* (21 %) constitute the other most problematic factors.
- Steepness is perceived as a problematic factor among relatively few respondents.




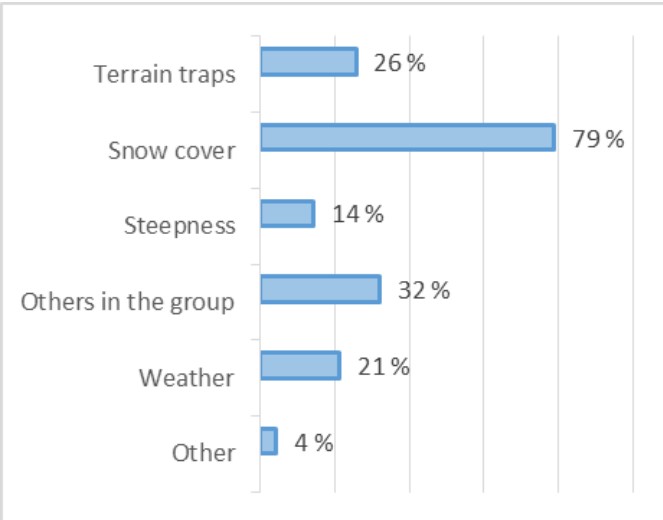

**Figure 4.** Factors recreational users considered difficult to assess and manage in order to have a safe trip in avalanche terrain.

### 3.2.2 Avalanche risk factors considered most and least important

In order to find out what the users consider as being the *most important element in the warning*, we asked "Which elements in
the avalanche warning are most important?". The respondents could choose multiple answers. Alternatives and results are
presented in Fig. 5. The results show that the participants perceive a relatively wide range of elements in the warning to be
important.

- A majority of recreational participants state that the *avalanche assessment* (68 %), the *avalanche problems* (65 %)
  and the *main message* (62 %) constitute the three most important elements in the warning,

- About half of the recreational respondents consider the *snow and avalanche analysis* (56 %) and the *danger level*
  (48 %) as important.

- Over a third of recreational respondents consider *snow and avalanche observations* (37 %), *mountain weather* (40
  %) and *management advice* (41 %) as important.

The results from the *expert respondents* were similar, with *avalanche problems* rated by far the most important factor (79 %).
We find no evidence that the elements chosen as most important depended on age, gender or experience (linear regression $R^2$
= .022, p = .224).





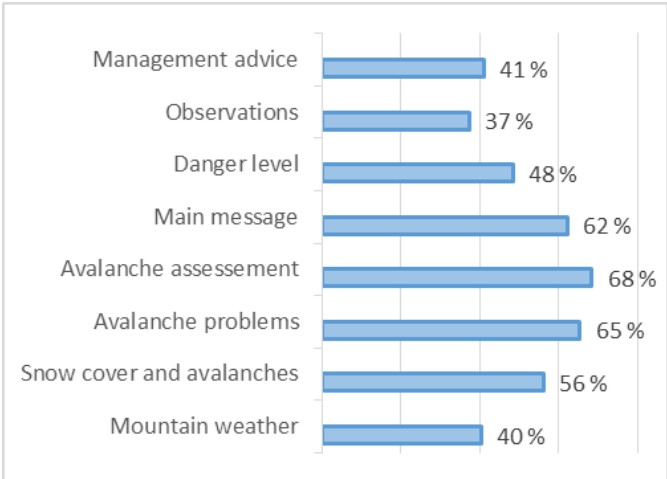

**Figure 5.** Factors considered most important in the avalanche warning on Varsom.no.

In order to find out what the users consider of *least importance or use*, we asked "Was anything of little use or importance? You may elaborate on the problem being format, content or other." A total of 69 participants responded to this question. 20 of these provided positive or neutral comments. We summarize the critical feedback, and our interpretation of this feedback, below.

- Seven respondents stated that they found *the mountain weather to be superfluous*, and that they rather used the standard weather forecast. Thus, clarification in the difference between the weather forecast and the summary of the mountain weather, and the link between the mountain weather history and forecast, and the avalanche forecast, is recommended.

- Five respondents stated that the warning contained *too many, and complex details and information*. These participants were mainly novices. This may imply that users with less skills and interest in the topic fail to get the key messages.

- However, another set of six respondents considered the level of detail as *too low*. These participants stated that the usefulness of the warning would be higher if it were less general, and if the forecast region was smaller. These answers point to the possibility that general forecasts for relatively large regions reduce the attention paid to the warnings.

- Three participants found the *core zone sector diagram* to be problematic. More specifically, these participants found it difficult to know if dark sectors represent safe, or unsafe regions. Although only three participants commented on this, their feedback is important since it implies that some users of NAWS may chose the unsafe sector because they misunderstand the graphics.

- Finally, four respondents found the *snow and avalanche observations* sometimes be *too complicated* or described in *too difficult terms*.



### 3.2.3 Elements easily misunderstood or poorly communicated

A total of 95 participants provided comments on if the avalanche warning contains parts that are *easily misunderstood or poorly communicated*. 30 of the comments were positive or neutral. We summarize the critical feedback, and our interpretation of the comments, below.

- 11 recreational respondents found the *core zone sector diagram* to be easily misunderstood. Like in the case of participants who stated that the core sector diagram to be of little use, these participants stated that they found it difficult to know which of the sectors (dark or light) that are most dangerous. Some participants suggested to add a legend or use more or different colours. These findings corroborate the findings in 3.2.2.

- Another 11 recreational respondents perceived that the regional warnings provide *too little details in terms of*
10
  *spatial or temporal variability*, and that the *forecasted regions were too large*. These findings corroborates the findings in 3.2.2.

- Eight recreational respondents found it difficult to understand the *danger level*, in terms of the meaning and consequence of it for the user. This is important, because if users do not understand the meaning of the danger level, they are poorly equipped to manage their risk exposure.

- Finally, six recreational users stated that the large amount of information provided in the warning made it difficult, especially for beginners, to decipher the key message.

The answers from the expert survey suggest that experts perceive similar factors (to be problematic as recreational users do: i.e., the *core sector and elevation diagrams*, *spatial and temporal variability*, the *danger level*, and *uncertainty*). However, the experts also pointed to a few problematic factors not mentioned by the recreational users: *Size* (especially the name "small" used for size 2), *probability* and *distribution*.

### 3.2.4 Missing information and features

In the final part of section B, we asked the respondents to identify missing information in the avalanche warning. 67 respondents provided comments. About 20 of these stated that no important information was missing. The elements asked for by the remaining 47 participants were the following:

- Observed weather and snow, and links to more detailed observations

- ATES recommendations (Avalanche Terrain Exposure Scale is a method for classifying the degree of terrain avalanche-exposure, Statham, 2012)

- Advice connected to competence levels, and

- More detailed warnings/information. Better visualisation of important weak layers (depth, type, etc)

We also asked the participants if some information or features are missing in the RegObs application. 81 users responded to this question, of which about 35 responded they did not use the application or were indifferent. The recreational users found the following to be missing:



- Weather data,
- A possibility to enter and record snow profiles,
- A possibility to read the avalanche warning (at least the danger level and avalanche problems) in the application,
- An opportunity to track trips,
5
- A user-friendly interface,
- Access to avalanches and avalanche paths,
- Information about actual elevation in relation to the avalanche problem elevation range, and
- Easy access to the snow cover history and relevant recent snow profiles nearby.

The results from the *experts* suggested that these pieces of information are missing:

- More precise description of where in the region or terrain the avalanche problem is expected, and where the danger level is expected to be lower, and
- A better description of the uncertainty and local variability.

### 3.3 Testing of text versus symbols or pictures

In section C, we asked the respondents to rate how well text, icons, and pictures communicate the avalanche problem on a

15 scale from 1= poor, to 3 = good. Each respondent evaluated two types of avalanche problems: a wind slab, and a persistent slab (see Fig. 6). A total of 222 recreational respondents completed this section of the survey.

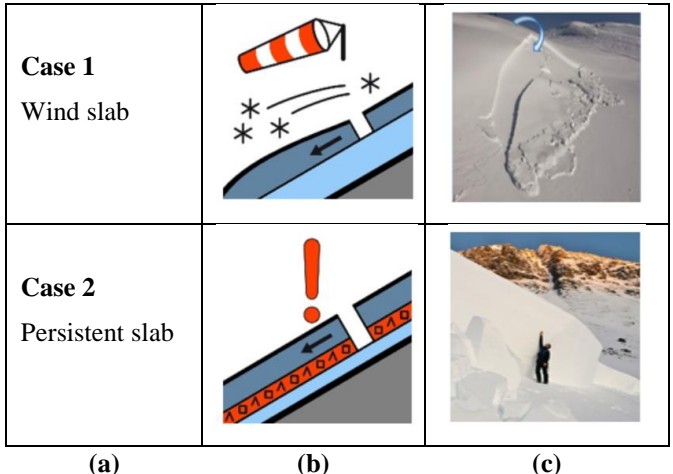

**Figure 6.** Test of what communicates the avalanche problem best in the avalanche warning: (a) text, (b) symbols or (c) pictures.

The results show that users preferred text and symbols to pictures (Table 5). 89 % rated the new EAWS symbols as good or

20 OK. The users were familiar with the names of the avalanche problems, which have been presented as text on Varsom.no during the previous three seasons. The users were less familiar with the symbols, as they were introduced at Varsom.no for the





2017/2018-season after becoming introduced as an EAWS standard in June 2017 (EAWS, 2017). Pictures have not been used in the warning on Varsom.no, but a few users may have seen the pictures on the avalanche school at Varsom.no. Notably, we found that the symbols were rated more positively the more experienced a respondent was, $\chi^2_{203} = 15.26$, p = .018. The text and pictures were rated equally irrespective of one's experience, p =.338 and p = .543, respectively.

**Table 5.** Results from test of what communicates the avalanche problem best in the avalanche warning.

| Text | Symbol | Picture | Rating |
|------|--------|---------|--------|
| 46 % | 51 % | 38 % | Good |
| 40 % | 38 % | 36 % | OK |
| 14 % | 11 % | 25 % | Poor |

**3.4 Testing of comprehension of the two scenarios**

A total of 177 recreational respondents completed the test for comprehension in Section D, by responding to one of the four
10   alternatives for each of the two scenarios. To recap, we asked the respondents to 1) rate how well they perceived that the avalanche danger was communicated, 2) what the most important behavioral implications of the warning was, and 3) what advise they would give to others based on the warning message. We measured how well the forecast persuaded the warning on a scale from 1 to 10. Of those who provided answers to this question, 21 % gave a rating of 10, and 56 % a rating of 8 or higher. Only 14 % gave a rating of 4 or lower. Mean ratings for the two scenarios (danger level 2, and level 4), and for each
15   of the four alternatives are presented in Fig. 7 below (left column). Fig. 7 also depicts the comprehension scores (right column). Higher scores indicate a higher match between the behavioral implications chosen by recreational users and experts. For the danger level 2-scenario (level 4-scenario) the minimum score is -4 (-6) and maximum score is + 4 (+3).





**Figure 7: (a)** User rating (left) and comprehension score (right) for the level 2 wind slab problem, **(b)** user rating and comprehension score for the level 2 wet slab problem.





We next compared the user ranking and comprehension score in more detail, by a) comparing the comprehension score to a score of 0, and b) investigating whether user ranking or comprehension differs between the four alternatives with ANCOVA's where a user's experience was a covariate. For statistical analysis we used JASP (JASP, 2018).

### 3.4.1 Danger level 2 wind slab problem

For the danger level 2 wind slab problem, the average user ranking of the four alternatives ranged from 5.1 to 7.4. The four alternatives were rated differently, $F(3, 172) = 10.124$, $p < .001$, $\eta^2 = .149$. Alternative 2, i.e. avalanche problem with technical information and advice, was rated highest and alternative 1, danger level with explanation as least informative. Post-hoc Tukey test confirmed it, alternative 1 was rated lower than the other three alternatives ($p$'s $= < .001$, smallest effect size Cohen's $d = .814$). A user's competence had no effect on the ranking of the alternatives, $F(1, 172) = .1966$, $p = .163$, , $\eta^2 = .010$.

Comprehension was good, with all four alternatives yielding overall positive scores, i.e. one-sampled tests for all four alternatives were significantly different from a score of 0 (alternative 1: $p = .015$, alternative 2-4: $p$'s $<.001$). Still, the comprehension scores were different for the four alternatives, $F(3, 172) = 8.188$, $p < .001$, $\eta^2 = .120$. Alternative 1, danger level with explanation had the lowest comprehension score and was significantly different from the other three alternatives, post-hoc Tukey tests had smallest $p = .021$, smallest effect size Cohen's $d = .541$. Notably, the higher the competence the

better was the comprehension, $F(1, 172) = 7.777$, $p = .006$, $\eta^2 = .038$. Finally, there was a positive correlation between user ranking and comprehension, $\rho = .2$, $p = .008$, 95% CI [.054; .337].

### 3.4.2 Danger level 4 wet slab problem

For the danger level 4 scenario, the average user ranking of the four alternatives ranged from 6.7 to 7.7. All alternatives (danger level, avalanche problem) were ranked high, and there was no statistical significant difference, $F(3, 172) = 1.787$, $p = .151$, $\eta^2$

$= .030$. Also user's experience did not influence the ranking, $F < 1$.
Comprehension was good, with all four presentations of the scenario yielding overall positive scores. All but alternative 3 had a significant positive score (alternative 3 only marginally: $p = .065$, Cohen's $d = .262$). The four alternatives did differ, $F(3, 173) = 4.188$, $p = .007$, $\eta^2 = .067$. Alternative 3, avalanche problem with technical information only, received a significantly lower comprehension score than alternative 1 (post-hoc Tukey: $p = .037$, Cohen's $d = .607$), and alternative 4 (post-hoc Tukey

$p = .006$, Cohen's $d = .763$). A user's competence had no effect on comprehension, $F(1, 172) = .93$, $p = .336$, $\eta^2 = .005$. There was also no relationship between user ranking and comprehension, $\rho = -.095$, $p = .207$.

### 3.4.3 Comparison

The results show that for the danger level 2 scenario, the three alternatives with the avalanche problems communicate more effectively than the one with the danger level. The user ranking and the calculated comprehension score provide consistent

results. For the danger level 4 scenario, on the other hand, the alternatives with the danger level and avalanche problem with travel advice score higher than the two other alternatives. The difference is clearer for the calculated comprehension score than



for the user ranking. The alternative with the avalanche problem and technical details seems to communicate least effectively however, it is also possible that users become too careful / conservative and rate factors as important that experts do not, and hence receive a lower score.

## 4 Discussion

The purpose of this paper was to evaluate how well the avalanche warnings, as provided by NAWS, are communicated to the public, and if some modes of presenting the warnings are more effective than others in communicating the *intended* message. We discuss our main findings below: Firstly, we discuss the user results in section 3.2 (avalanche warning), secondly we discuss the results in section 3.3 (modes of communication: text versus symbols and pictures), and finally the results from the test of comprehension reported in section 3.4.

**4.1 The avalanche warning**

Our survey responses suggest that users find it most difficult to assess and manage the *snow cover,* but that relatively many individuals also find it challenging to manage *group dynamics*, *terrain traps* and the *weather*. The fact that so many (both experts, experienced recreational users and novices) struggle with an evaluation of the snow cover and its impact on the avalanche hazard is not surprising. The snow cover *is* difficult to assess and manage, it can vary considerably over both short

time periods and distances. It is created by a complex and dynamic interaction between the atmosphere, the old snow cover, and the ground, and the development of the snow cover over time may create complex structures and properties. Visual surface clues are few and information on the internal structure and properties hard to come by. The difficulty in assessing and managing the snow cover is reflected in what users perceive as most important in the avalanche warning: the *avalanche assessment*, the *avalanche problems* and the *main message*.

A well-functioning avalanche warning needs to translate the complex dynamics and characteristics of the snow cover and avalanche hazard into a clear message that novices, recreational users and experts find useful and can translate into behavioral implications. However, although the responses to our survey questions suggest that both expert and recreational users demand this type of information, the responses also show that it is challenging to create a warning that will fit all needs. While some users would like to see more detailed information on the type of avalanche problems and geographical distribution of these

problems, the characteristics of the snow cover (including spatial and temporal variability), weather patterns, and estimates of uncertainty in the forecast, other users state that the amount of information and detail currently available in the warning is already too high and complex, and that it makes them confused.

Today, the Norwegian avalanche warning describes the avalanche hazard using both symbolic representations, a summary of the behavioral implications, and more lengthy descriptions of the avalanche problem, the snow cover, and the mountain

weather. In addition, users have access to snow observations via RegObs. The symbolic representation of the avalanche danger



and main message is currently presented at the top of the page, while more detailed information is available lower down or to the side.

Our interpretation of the responses given in the survey, by both novices, recreational users and experts, is that the avalanche warning should at large maintain its current structure, with easy-to-grasp information for all users (novices as well as experts)

at the top level of the warning, and more detailed and complete information for advanced users, e.g. information about the type of avalanche problem, character, timing, geographical distribution, and reliability of observations, at a lower level. On the other side, most experts (79 %) and recreational users (65 %) rated the avalanche problem as the most important element of the warning. This, in combination with several users saying that there is too much and too complex text (i.e. redundancy), suggest that the avalanche problems should be communicated high up in the warning. A more compact presentation with less

information would strengthen the communication efficiency, in particular if the overlap with the avalanche danger assessment text is reduced. We also see several other areas for improvement. 1) The *danger rating* was considered important by many, but by less than 50 % of the users. This may suggest that this element may be better off at a less pronounced place on the page. 2) Many individuals find it difficult to interpret the *core sector* and *elevation diagrams,* mainly in terms of identifying safe and unsafe sectors. To remedy this problem, it may prove beneficial to show the danger rating and/or avalanche problem in

two-three different elevation bands, as is done by several European and Canadian AWS', or to use bold red colours and clearer fonts. 3) Information on weather is repeated in several places in the avalanche warning, e.g., in the weather forecast, the avalanche problems and the snow cover discussion. To improve clarity and readability, it may prove beneficial to remove redundant information about the weather. 4) To increase the usefulness of snow observations, the interface of RegObs may need revision.

**4.2 Modes of communication: Text versus symbols and pictures**

Our empirical analysis shows that most respondents prefer symbols and text to pictures. The preference for text messages may partly be explained by the fact that users have become accustomed with this mode of communication: NAWS have presented the avalanche warnings using text during its five years of operation. Another potential explanation is that the names of the avalanche problems are easily communicated verbally – in interviews in the media, during avalanche courses and when

discussing the avalanche danger before and during trips. Even though symbols are efficient, text labels are very useful. The EAWS symbols were new to Norwegian users, and the positive rating of these supports the decision to introduce these symbols as a standard in Europe. Unfortunately, it was beyond the scope of this study to test if the wording of text messages and the symbolic representation of avalanche problems are optimal for persuading the intended message. To do this, more advanced testing is needed.

The relatively low rating of pictures may be explained by the choice of certain pictures, or by the amount of details in them. Pictures are bound to be taken at a certain location, under certain circumstances, and almost always contain some amount of irrelevant information. As a consequence, the main message may become blurred, and some users may feel that the snow or landscape is not representative for the avalanche forecast or region. Further tests of pictures, or even video, could be carried



out to explore if these, otherwise very effective media, could be used to communicate the avalanche danger, perhaps as a complement to the other parts of the forecast. NAWS has posted pictures and videos of the current situation a number of times on Facebook, and especially the videos have had a large impact measured by the number of views, likes, comments and shares. This suggests that a more elaborate study on the effect of pictures and videos should be carried out. However, the costs of obtaining relevant quality video and pictures may outweigh the benefits.

## 4.3 Modes of communication: Comprehension

We evaluated comprehension of the communicated avalanche warnings by the use of two methods: we first asked respondents to rate how well the avalanche hazard and the associated behavioral implications were communicated, and thereafter tested if different modes of communication resulted in different "comprehension scores". All participants evaluated a danger level 2-scenario (wind slab), and a danger 4-scenario (wet slab). In each of these scenarios, the participants were randomly exposed to one out of four alternative descriptions of the avalanche hazard and asked to choose the behavioral implications associated with the warning. The comprehension score was based on a comparison between the choices made by recreational users, and a template constructed from the answers made by a panel of avalanche experts.

Our empirical analysis of the subjective rating of the avalanche warnings show that most recreational users perceive that Varsom.no communicates the avalanche hazard in a good way: 51 % of the users rated the communication of the danger in the level 4 scenarios as 8 or higher, on a scale from 1 to 10 (41 % in the level 2 scenario). These results are consistent with previous studies on user satisfaction (Kosberg et al., 2013, Barfod, et al., 2014, Barfod, et al., 2015) and the conclusions in a recent evaluation of NAWS (Hisdal et al., 2017).

The highest rating dependent on the scenario. For the level 2 (wind slab) scenario, the alternatives 2 (avalanche problem and technical information and advice) received the highest rating, slightly higher than alternative 4 (avalanche problem and advice). Alternative 1 (danger level) was rated the lowest. For the level 4 (wet slab) scenario, it was alternative 1 that received the highest rating but this was not different from any of the other alternatives. These results suggest that the users perceive that they need more detailed information than just a danger rating when given a danger level 2, but are highly satisfied with knowing the danger level if it is 4. Notably, for danger level 2 a user's competence mattered when it came to the rating of the alternatives, but not for danger level 4. This suggests that the value of more detailed information about the avalanche problem increase as the user's competence level increase. Level 4 might be a cut-off for most, in terms of making the decision not to enter avalanche terrain. The avalanche-related demographics data from the user respondents (Section A and E) showed that the more competent the users, the more ski tours they undertake. However, quite a few of those without competence or courses are also active ski touring. Most respondents assessed themselves as being competent. Hallandvik et al. (2017) showed that novices assessed the terrain for a specific site as less complex than experts, they weighted information in the avalanche forecast differently, and used different strategies to gather information about the snowpack on a trip. Thus experience and competence matters to a certain degree when communicating avalanche danger.



Our results from the test of the level of comprehension for the level 2-scenario (wind slab) are at large consistent with participants' subjective evaluations: participants score significantly lower on the comprehension score if the only information available is a danger level with a standard explanation (alternative 1). We therefore argue that a simple danger rating is not enough to convey the intended warning message on lower danger levels. At this level, the users are considering how to travel

in avalanche terrain, rather than whether or not to enter avalanche terrain. Rather, in this situation the warning should present the avalanche problem with a reasonable level of details. Our results do not provide a clear answer to the question of which details that are most important in order to communicate the message, indeed all three alternatives yielded a higher comprehension score than alternative 1.

The results from level 4-scenario (wet slab) are markedly different from the level-2 scenario, both in terms of comprehension
scores, and in terms of the match between objective comprehension and subjective evaluations. Opposite to the level-2 scenario, users rated all alternatives equally. However, in this scenario, the comprehension scores was significantly *lower* for alternative 3 than it was for alternative 1 and 4. In other words, leaving out the advice and explanation resulted in a lower comprehension. Note though that all four alternatives yielded positive scores, most users did select the same factors as did the experts. One possible explanation to the observed lower comprehension score for alternative 3 is that the technical information

caused confusion rather than helped the respondents.

Based on the results of our empirical analysis our recommendation is that NAWS should carefully assess the importance and priority of details presented with the avalanche problem in order to shorten and simplify the communication of the danger to the different user groups. It is important to consider the redundancy of information. If two or three avalanche problems are presented in the warning, the redundancy between the details in the problems as well as the main message and danger

assessment may be considerable. NAWS should consider how to normalize the avalanche warning, in order to avoid repeating the same information several times. Our results also points to the importance of communicating the avalanche problems at a high level in the warning, especially for warnings on lower danger levels. However, both the avalanche situation and the snow cover may be very complex across the warning region and forecasting period. The NAWS warning regions are about 20 times the European average (Engeset 2013, Techel et al., 2018), and some have rather complex topography and weather patterns.

Thus, simplicity may not always be achievable, without sacrificing important information for the users.

There are several factors that may have affected to the results presented in this study, and we would like to linger around them. First, it is important to note that this study targeted recreational users, and not preparedness authority personnel. The needs of novice, advanced, and professional users are likely to differ. While simple symbols and messages are likely beneficial for individuals with less avalanche knowledge, technical details can be of great use for advanced users. Properties such as sector,

elevation, size and probability of release are useful when considering which roads or residential areas are exposed to the avalanche danger. We therefore recommend that a more detailed study should be carried out, in order to investigate what is effective to communicate to preparedness users. Second, the Norwegian users were only recently introduced to a national avalanche warning system. This implies that many users have been in the process of acquiring knowledge about how to assess avalanche danger during the period from 2013 to 2018. Other than the avalanche warnings provided by NAWS, few sources





for assessing the avalanche danger are available in Norway. Most avalanche courses frequently use the avalanche warnings and avalanche educational resources on Varsom.no. Finally, we only tested a limited set of alternative communication modes to the participants. To fully evaluate how to optimally design the avalanche warning, tests with more variations and scenarios are needed.

**5 Conclusions and recommendations**

In this study, we investigated how well recreational users perceive that the avalanche warnings on Varsom.no (www.varsom.no) are communicated, and if different modes of communication affect the level comprehension of the hazard at hand. We also identified elements in the avalanche warnings that recreational users perceive to be of greater or lesser importance, are easily misunderstood, or missing.

Based on our empirical analysis of the data, we make the following conclusions and recommendations:

1. *Redesign core zone and elevations graphics/text.* Problem: Participants found it difficult to understand if the avalanche problems were present or absent in coloured sectors. Possible solution: show the danger rating and/or avalanche problem in two-three different elevation bands, as is done in by several European AWS' and in Canada.

2. *Less is more.* Problem: the amount of text and details in the warning reduced the motivation to read the warning and

made it more difficult for the user to pick up the main message. Possible solution: Minimize repetitive information and reduce complexity.

3. *Local information matters.* Problem: the avalanche warnings are produced for relatively large geographical areas with big spatial variations in the snow+ cover. Possible solution: use maps to show the parts of the region most affected by the avalanche problem(s), or where the avalanche hazard is expected to be higher (lower). Another way

could be to present local weather history, and/or snow observations from automatic stations, or to present the snow history by visualising some manual snow observations as time series.

4. *We need to teach (1)!* Problem: a very large share of both expert and recreational respondents state that they find it most difficult to assess and manage the snow cover. Possible solution: present the avalanche problem, snow cover analysis and the avalanche danger assessment in a more systematic and pedagogical manner in order to improve the

competence of the users. It should be noted that even the experts considered the snow cover as the most difficult factor, suggesting that it is complex to manage for users at all levels.

5. *We need to teach (2)!* Problem: A relatively large share (more than a third) find it difficult to identify terrain traps, and to manage others in the group. Possible solution: Use the "avalanche school" to educate users about terrain traps and talk about group dynamics to help users make better choices about who they choose to recreate with in

avalanche terrain.

6. *Avalanche problem is important.* Problem: the avalanche danger level is not enough for making decisions in avalanche terrain, more detailed information is needed. Possible solution: Promote the avalanche problem,





especially at danger levels 2 and 3, which also are the conditions under most fatalities occur. Streamline the presentation of the avalanche problem according to danger level and reduce overlap with the avalanche danger assessment in order to reduce the complexity for users. Reduce the amount of information to recreational users at higher danger levels. The danger level was rated as important, but somewhat difficult to understand. It is a simple

numeric value, but is determined from relatively complex and subjective factors, and is probably difficult for users to understand and use.

7.  *Keep the EAWS symbols.* The users considered the new EAWS standard icons for the avalanche problems to communicate the danger well, although the users were not familiar with the icons.

In conclusion, our study has confirmed that the communication of the avalanche danger on Varsom.no is perceived as effective

by the users. The results of the testing of the effectiveness of different alternatives for communication of level 2 and level 4 avalanche danger suggested that the avalanche problems communicated more effectively than the danger level at lower danger levels. At the higher danger levels, no significant difference was found between the alternatives. The results suggest that a simple danger rating is not enough to convey the intended warning message on lower danger levels, rather the warning should present the avalanche problem with a reasonable level of details. At higher danger levels, the results suggest that leaving out

the advice and explanation resulted in a lower comprehension. For danger level 2 a user's competence mattered when it came to the rating of the alternatives, but not for danger level 4. Most experts (79 %) and many recreational users (65 %) rated the avalanche problem as the most important element of the warning.

Based on the findings in this study, NAWS redesigned the avalanche warning on Varsom.no: the communication of the core sectors was improved (displayed in red signal colour rather that vague grey), a location search function was added, the display

of avalanche problems was moved up to just below the main message, the redundancy in information between the avalanche problem, snow cover analysis and the avalanche danger assessment was reduced, the region map down to the bottom of the page and the and mountain weather and snow cover analysis were restructured.

Norwegian users, experts and avalanche warnings were used in this study, but we believe the methods and results are important to the wider scientific community and the AWS' in other countries. The building blocks and communication techniques of the

avalanche warnings on Varsom.no follows the standards of EAWS and Varsom.no and NAWS was developed in collaboration with a number of AWS' in Europe and North America.

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
