# Peer review of "Communicating public avalanche warnings – what works?"

_Natural Hazards and Earth System Sciences, 2018_

## Referee Comment (RC1) · Anonymous Referee #1 · 8 Jul 2018

I have reviewed the manuscript "Communicating public avalanche warnings – what works?" and offer the following comments:

The subject matter of this manuscript important and offers a solid contribution to the avalanche literature as many public avalanche warnings have not been tested. The topic is appropriate for publication in NHESS.

Overall I found the writing quality to be good, but at times wordy and difficult to interpret what the author was trying to say. I found I had to re-read significant portions of the manuscript, and in some cases I remain uncertain if I understand what the authors are getting at.

I have two overriding concerns with the paper, which might be attributed to my inability

to understand the writing (rather than the method), but I believe the paper needs more clarity on the following:

1. The use of the Expert Survey. It is described that the survey was given first to a group of experts in order to derive a template of "correct" answers. Further on it is described that the experts answers were used to establish a "communications effectiveness score". I remain unable to understand how the opinion of the experts should be/was used as the correct answers with which to compare recreational or novice users? A more thorough explanation of the relationship between the two survey groups, and why the expert's answers are suitable for being an answer template is necessary.

2. Comprehension testing. The testing method for comprehension does not seem very robust. Comprehension of an avalanche warning should be demonstrated by specific actions on the ground, in the terrain. People need to be able to say "where" the avalanche warning applies, and where it doesn't in order to demonstrate comprehension and this requires being specific. They need to be able to make choices about which trips/slopes they will do, and which trips/slopes they will avoid. The 9 questions posed (table 3) do not seem specific enough to infer comprehension of an avalanche warning. My impression is that the questions asked are not sufficient for making any conclusive statements about people's comprehension of an avalanche warning.

Following are additional comments:

1. P1 Line 20 – what is indented comprehension? 2. P2 Line 27 – cite a reference for the 100 km2 statement 3. P2 Line 28 – Jamieson et al 2008 is not listed in the references 4. P3 Line 7 – add "locations" as part of recently observed avalanches in the region 5. P4 line 10 – I question calling avalanches a "low probability phenomena". Particularly in relation to other natural hazards, avalanches have an annual return period and many locations release multiple times per winter. I do not consider this low probability. 6. Figures 1, 2 & 3 – these figures are mostly unreadable. I recommend

making higher quality figures where the subject of the figures can actually be read. Currently only the general layout of the web screen is available from these figures. No detail can be read, yet this detail is essential to see the product that is being tested. 7. P4 Line 30 – typo halter? Should this say "halt" 8. Figure 2 – consider breaking this into 3 figures so that it can be read. 9. P8 Line 12 – I do not think RegObs is the only open-access online real-time distribution system for avalanche forecasting (see MIN, Avanet). 10. P11 Line 9 – Section D in Table 1. Typo? No section D in table 1. 11. Figure 3 – same as comment on Figure 2 – currently the details of this figure are not readable, and they need to be since they are the basis for the testing. Break into several figures. 12. Table 3 – I cannot find an explanation of the "4-scenario response". This needs to be clear as its not clear from just the table alone. 13. P12/13 – the description of how the communication effectiveness score was obtained is not clear. Despite reading several times, I remain unsure if I understand this. Ensure this method is explained well. 14. P26 Lines 22/26 – We need to teach (1) is a poor header that does not communicate.

In Summary

- The manuscript is good and I recommend it be published after minor revisions are completed - Needs better figures. Current figures 1,2 3 are poor and unreadable - Overall the writing could be streamlined to improve comprehension. - Better explanation of how the expert survey was used as an answer template plus a defence of why the experts are the ones to measure against. - Comprehension testing methods seem questionable – need a better explanation

And finally – the concept of avalanche terrain is lacking throughout this manuscript which is understandable because the goal was to test the NAWS product. However, interpreting an avalanche warning and putting it on the ground in avalanche terrain is fundamental to comprehension of an avalanche warning and the lack of discussion regarding avalanche terrain stood out for me as I read this paper. For example, the genesis of avalanche problems was because different problems manifest in different

places in the terrain. True comprehension of an avalanche problem would be under-standing where the problem does and does not exist. I was always wondering, where is the terrain part? This paper does not demonstrate the ability of anyone to read an avalanche warning and then put their understanding to work in the field.

———————————————

---

## Referee Comment (RC2) · B. Zweifel (Referee) · 17 Jul 2018

**General Comments**

**Overall summary**

I enjoyed reading this paper. As an avalanche forecaster I'm personally very interested in improving our communication. I think this study contributes significantly to do so. Some of the recommendations as given at the end of the paper should find its way back into the ongoing discussion within EAWS.

My recommendation to the editors is to publish this article under reserve of the revisions as suggested below.

Since I and many other avalanche experts assume the communications design of the avalanche reports at least as important as theirs content, I think you addressed a very important topic with this study. Your survey was very well designed and organized; accordingly your results are based on a sound data basis – one of the main strength of this paper.

I have two major concerns with the paper: (1) The weights you establish for a communication effectiveness score are on the one hand difficult to understand and on the other hand it is hard to understand the reason behind the chosen percentage classes. (2) In the result section I would prefer a diagram illustrating both the answers of the experts and the answers of the larger group. In my eyes this would make it easier to understand.

Since I see this piece of work as very relevant I encourage the authors to undertake the suggested revisions.

**Specific Comments**

**Introduction**

The introduction in the topic provides the necessary background to get into the topic. I have only minor comments:

Page 2, Line 33: Instead of Landrø et al., 2013 I would rather refer to EAWS, 2017 which would be the newer synthesis.

Page 3, Line 2-4: Travel advises are in some regions also given within the avalanche danger assessment section of the report. There they do not resist as a general advice but more as a specific advice for the actual situation.

**Methods and data collection**

Either in this section or in the discussion at the end I encourage you to address a possible bias of your study sample towards participants with an over-average interest in avalanche safety and its possible different answers (e.g. Haegeli, Strong-Cvetich, and Haider (2012), p. 806).

Page 11, 12: Is there any reason how you determine the classes for positive and negative weights of the communication effectiveness score? For my feeling this was one of the weak points of the analysis. It seems a bit made up out of thin air. Maybe you can bring some more evidence how you chose this classes?

Page 13, Line 18, 19: I guess that this values could be biased towards a population with more avalanche expertise than average (see point described above).

Page 12, Line 28, 29: As well here a see a possible bias towards people who use the forecast more often than average (see point described above).

Page 14, Table 4: "Ski tours per year" is in my opinion only one important value of experience. It is also important to know how many years of experience someone has. Did you survey this as well? Further, I suggest adding total values in Table 4, so that one can see how many percent of all members fell within the different competence classes.

**Results**

I was a bit confused to find answer values from the experts here in the result section since I understood from the method section (page 11, line 4) that the expert answers just define the "correct" answers. Anyway I recognized that the comparison between the two values (from the experts and from the recreational users) is of interest of course. Therefore I propose to adopt Figure 4 and 5 in a way that you show both values (from the experts and from the recreational users) for each factor.

Further I have some minor suggestions:

Page 17, Line 15, 16: I assume that this point corresponds to paragraph 3.2.2 as well? Maybe you mention that here.

Page 17, Line 19: I would write "Avalanche size" instead of "Size" only, so it gets clearer.

Page 18, Line 1-8: This is quite a bit of criticism and would demand major changes in the RegObs application. It does not really become clear, whether you intend to do so or not. I suggest you to give this point some more weight in the discussion section.

Page 18, Figure 6: In my opinion the pictures are not really very suitable for what you want to show. The wind slab could also be a persistent slab or a new snow problem. A picture with more varying slab thickness would be clearer in my eyes. Further, the picture showing the persistent slab shows only the upper fracture. This could also easily be from a wind slab problem. You address this problem in the discussion section anyway, so I see not really a need for action here.

Page 19, Line 16, 17: I would prefer to have two sentences here (instead of the parentheses), it would make the text more readable.

**Discussion**

The discussion is interesting and addresses the general points. However, in some points I suggest to go a bit more into detail:

Page 22, Line 15: Probably you could add a reference here which addresses the of the snow cover (e.g., Schweizer, Kronholm, Jamieson, and Birkeland, 2008).

Page 23, Line 15: Would you have any examples of AWS who show different danger levels and/or avalanche problems and is it possible to shortly discuss pros and cons of it?

Page 23, Line 15: Isn't there any possible conflict with red color for the core zone with the red color from danger level 4?

Page 23, Line 21: A mentioned above I see the pictures as not perfectly suitable for the phenomena you want to show. This means that your chosen sample is probably not the best and I find it delicate to generalize that pictures are not suitable for this kind of communication. However, I saw that you addressed this point properly later on (line 30 ff) and accordingly do not see a necessary action here.

Page 24, Line 27, 28: Maybe you take my comment from above with the number of years in experience into account here.

**Conclusions and recommendations**

Some minor points could be more precise here:

Page 26, Line 4, 5: How would you illustrate parts of regions most affected by an avalanche problem? Would you further divine the region into sub-regions? This does not become clear here.

Page 26, Line 4, 5: What do you mean with higher or lower avalanche hazard? Are you talking about another danger level or about variations within one danger level?

Page 27, Line 5, 6: I suppose here to refer to a simple "public danger scale" which has only a short characterization and a travel advice for backcountry recreationists for each danger level (e.g., https://www.slf.ch/en/avalanche-bulletin-and-snow-situation/about-the-avalanche-bulletin/danger-levels.html, https://avalanche.org/avalanche-encyclopedia/danger-scale/).

Page 27, Line 19: Concerning the communication of the core zones it would probably be worth to develop a EAWS standard design. However, I know this will be a challenge. I don't think you have to mention that in the paper anyway.

**Technical note**

Page 3, Line 17: There is a point missing after "(DeJoy, 1999)"

**References**

Haegeli, P., Strong-Cvetich, L., & Haider, W. (2012). How mountain snowmobilers adjust their riding preferences in response to avalanche hazard information available at different stages of backcountry trips *International Snow Science Workshop 2012, Proceedings* (pp. 800-806). Anchorage, AK.

Schweizer, J., Kronholm, K., Jamieson, J. B., & Birkeland, K. W. (2008). Review of spatial variability of snowpack properties and its importance for avalanche formation. *Cold Regions Science and Technology, 51*(2–3), 253-272. doi:http://dx.doi.org/10.1016/j.coldregions.2007.04.009

---

## Referee Comment (RC3) · M. Staples (Referee) · 10 Aug 2018

*My apologies if these comments are difficult to understand. I have been working on several emergency wildfire assignments in Utah and Oregon and have been adding comments to this document when I've had a few spare minutes here and there.

Communicating public avalanche warnings – what works?

Overall this seems like a great study and a great effort to quantify the effectiveness of different elements of avalanche warnings. It's a very difficult topic, and I think they did a great job combining qualitative and quantitative data to form a picture of what maybe works and what doesn't. The topic is very significant scientifically as we continually debate the format, style, layout, etc of avalanche warnings; however, we have very

little data on how well people understand our products. Part of this study collected is what people like and what they want. What our users prefer may not be the same as what helps them understand the avalanche hazard, and what causes them to alter their behavior. The section on comprehension was useful. This does not conclusively answer many questions, but it provides solid guidance for the NAWS to provide their users with a useful product.

My background includes running a large avalanche center in the U.S. as well as a graduate degree in engineering. I do not have a background in statistics, nor do I have any expertise in designing surveys. For these reasons, I am not qualified to comment on the scientific quality of the statistical part of this work.

Some of the biggest issues were with the presentation and writing. It was very difficult to follow this paper and understand what they did.

Section 1

The challenges of communicating avalanche hazard to the public (section 1.2) were described very well. However, the first part of this section (lines 15-26) about impact-based warnings was confusing. If exposure and vulnerability are determined by individual users, how can AWS's issue impact-based warnings?

U.S. successes (page 4, line 28) perhaps don't match other trends and could be worth mentioning. U.S system is unique in that it has very different styles and formats yet seems to be effective. The trend in the U.S. has been a declining fatality rate. The number of fatalities has been flat while use has surged, thus the rate has declined. https://avalanche.org/2016/06/27/2016627us-avalanche-fatality-trend-is-flat-for-the-past-22-seasons/

It would be very helpful to have an English version of Figure 2 (page 6). This would help some readers really understand the content of elements of avalanche warnings. Do any of the sections in the avalanche warning use stock language? Are they written

from scratch each day? This is extremely important to know. Whether or not it contains original writing has a major effect on credibility.

Many issues with this paper seem to stem from a translation issue. Use of some words like "danger" and "hazard" made this paper very confusing to follow. It was sometimes unclear when the word "danger" meant specific danger rating (i.e. Low, Moderate, etc.) or a general reference to hazard. Sometimes they called it the "danger rating" other times the "danger level". While not a huge issue, this made the paper confusing.

Table 1 (page 7) needs to be rewritten. This table is referenced in Figure 5 (page 16); however, the element names in Table 1 and Figure 5 do not match. They are not presented in the same order either. Both of these issues make it very difficult to understand the results of this study. This was a serious problem for me.

Figure 2 shows elements of the avalanche warning. Each one is labeled with a number, and the caption has a description for each element. Figure 5 shows how users ranked elements in the avalanche warning. However, it was difficult to cross-reference these two figures because the wording in the caption for Figure 2 does not exactly match the wording in Figure 5. Additionally, the elements in Figure 5 are listed in a different order than they are in Figure 2. These issues made it challenging for me to fully understand the results.

Section 2

The way the male/female demographic was described on page 9, line 10 should be changed to mirror the way it is described in page 9, line 20. Line 10 perpetuates gender biases.

Survey design: I do not know how to design a survey, but I know the way questions are phrased can have a big effect on responses. I assume that questions on this survey were written in a neutral way.

It seems very difficult to truly assess comprehension. Given this difficulty, the authors

did a great job trying to quantify comprehension with their system described on page 12, lines 1-5.

Why did you test comprehension of wet slabs? Do they kill a lot of people in Norway? Persistent slab avalanches kill many people in the U.S., and they are difficult for users to understand. Risk management and travel advice messages for persistent slab avalanches are difficult to communicate. Additionally, there can be significant message fatigue with this avalanche type. Personally, I would have tested comprehension regarding this avalanche type.

Wind slabs were a good problem to test because they are so common.

The four alternative ways to present the forecast (is it a forecast or is it an avalanche warning?) on page 11, lines 10-12 do not match items listed in Table 2, section D. For example, item 1 is listed in the table as "Avalanche danger with explanation (general advice associated with the danger level)" and it is listed on page 11 as "only the avalanche danger level and very limited travel advice". This is not a major issue, but it makes it hard for me to follow the paper. Is "general advice" the same thing as "very limited travel advice"?

The communication effectiveness score, page 13 line 1, seemed like a great way to assess the responses from participants. Is it perfect? Who knows? Using "expert" answers as a way to evaluate participant answers seems like a great process to me.

Section 3

Another inconsistency involved the level of avalanche knowledge. The categories mentioned on page 13, lines 18-21 do not match the categories of "competence" listed in Table 4 (page 14).

In table 4 the level of experience is categorized by "ski tours per year". How did you measure the experience of other users like snowmobilers, snow shoers, etc?

Again, the warning elements in figure 5 do not match the elements of the warning listed

in table 1 and figure 2. While the authors may be referring to the same elements, using different wording to describe the elements made it very difficult for me to understand.

The authors did a great job summarizing the qualitative results. In the few surveys I have conducted with users in the U.S., comments often contain the most valuable info. Sometimes a single comment from a single person can be the most valuable part of the survey.

Page 21, lines 9. The authors comment that "user's competence had no effect on the ranking" in line 9. Did experience have an effect? As I understand it, "competene" and "experience" are two different things.

Similarly, in line 20, the authors comment that "experience did not influence the ranking." What about competence? Later in line 25 they say that "compentence had no effect on comprehension". Which is it? Is it both? This is confusing and not clear.

Section 4

Page 22, lines 20-22 – That sentence states the purpose of an avalanche warning really well.

The discussion of symbols vs text, page 23 line 25, is interesting because the tech industry has struggled with and gone back and forth on. Symbols can be confusing. Text is not, but there are issues in translating between languages.

Page 24, line 15, The words danger and hazard seemed to have been used interchangeably. While they may mean the same thing, it would be better to pick one for this paper. It could be easier to use the word "danger" when referring to the danger level. This added an extra layer of confusion for me.

Page 24 lines 22-24 suggest that danger level is well understood by users. Section 3.2.3 suggests that users have difficulty understanding the danger level. Another inconsistency is that the authors would sometimes say "danger level" and other times say "danger rating". It would help to use just one.

All of page 25 is a great discussion. A huge issue is you addressed is when 2-3 avalanche problems are present. There are almost always 2 problems present. Great topic for further discussion in another study.

Section 5

This study delivered some concrete findings. It seems to have been well designed, but it was challenging to understand because of the writing. The authors did a good job with their conclusions by not presenting conclusions with too much specificity. It would be easy to read too much into the results, but they did a good job of keeping their conclusions more general.
* * *

---

## Author Comment (AC1) · 16 Aug 2018

**Open discussion** https://www.nat-hazards-earth-syst-sci-discuss.net/nhess-2018-183/#discussion

**Authors' response to RC1 review by anonymous referee.** The response is shown in blue.

**1. The use of expert survey**

In our study we aimed at testing to what degree the users comprehended the intended message of the avalanche warnings. In order to do this we needed to find out what is the intended message, as defined by the sender (NAWS). This was done by letting the NAWS personnel (forecasters and observers) complete the survey and in this way defining what they meant the avalanche warning should communicate.

After this was done, weights were allocated to the answers, as described in chapter 2.3. Then an open survey was conducted, where the users could give their responses, the user survey.

There are probably several ways to do test how well the warnings are communicated. The method applied in this study compares the message as understood by the users with the intended message by those producing and publishing the warning. The user survey include respondents with different levels of experience and competence, not only novices. In fact, the users are novices, intermediate and experts. However, they are all users. On the other hand, the expert survey was restricted to NAWS personnel.

We will clarify this in chapters 2.1.1 and 2.1.2 by explaining that NAWS personnel participated in the expert survey, while all types of users participated in the user survey. Participants in the expert survey were NAWS experts (personal invitation only) and participants in the user survey were users (open invitation, anyone could participate). User survey participants included all types of users (various degree of competence and experience, from beginners/novices to experts; various types of use, from recreational to professional and preparedness). Expert survey participants included only forecasters and observers in NAWS, all trained in the same system.

**2. Comprehension testing**

We agree that testing the understanding of the avalanche warning by analysing actions and decisions in the field would be very interesting. Indeed, this was our first approach. However, we also had to establish the efficiency of the warnings more generally, for which using a web-based survey is appropriate (as it is less complex and easier to eliminate factors that are not attributable to the communication of the warning). Each method has its advantages and disadvantages. Web-based surveys are cost and time efficient, show high response rates and ensure voluntarily responding as the user without any costs can abort the study. Importantly, anonymous web surveys compared to active tracking in the field has high data security, i.e. we do not need to ask people to carry recording devices on their trip or requesting data logs from apps, which allows identifying the user. This often leads also to more honest answers, as users "know" they are anonymous, which is not the case in fieldwork. Field-based testing also has a few challenges. There is a large body of evidence showing that there is a major difference between behavioural intentions/attitudes and actual behaviour. Therefore, if we would focus solely on the behaviour in the field and found a miss match between this and the communicated warning, we would not know where along the line from information to action things went wrong - if this was due to lack of comprehension or rather risk seeking attitude.

We also believe a web-based survey is relevant, as many decisions are made based on the avalanche warning ahead of getting into navigation in the terrain. It could be decisions such as to choose forest rather than the alpine for today's trip, or delay the planned trip a few days until the snow stabilises. As cited in our paper, users do use the avalanche warnings, so the first thing to do should be to reassure us that the warnings are understood by the receiver as the sender intended.

We will describe the value of using web-based testing in more details in Chapter 1.4. At the end of Chapter 5, we will write that we recommend a follow-up study with in-the-field-testing, where avalanche problems and terrain choices are given much more attention.

**Additional comments**

1. P1 Line 20 – what is indented comprehension?

It is the message NAWS intend the users to understand and pick up. We will replace "comprehension" with "message".

2. P2 Line 27 – cite a reference for the 100 km$^2$ statement.

We will add "EAWS 2017", the EAWS MoU at https://lawine.tirol.gv.at/data/eaws/MoU_EAWS.pdf.

3. P2 Line 28 – Jamieson et al 2008 is not listed in the references

Thanks, will be added.

4. P3 Line 7 – add "locations" as part of recently observed avalanches in the region

Will be added.

5. P4 line 10 – I question calling avalanches a "low probability phenomena". Particularly in relation to other natural hazards, avalanches have an annual return period and many locations release multiple times per winter. I do not consider this low probability.

We will rephrase this sentence (we found this at line 11), and explain that many user have never or seldom experienced a release of an avalanche themselves.

6. Figures 1, 2 & 3 – these figures are mostly unreadable. I recommend making higher quality figures where the subject of the figures can actually be read. Currently only the general layout of the web screen is available from these figures. No detail can be read, yet this detail is essential to see the product that is being tested.

The purpose of Fig. 1 was to show the forecasting regions, which are readable. We may remove the rest of the web-page capture. The purpose of Fig. 2 was to illustrate the layout and elements included in the warning, not the detailed text. The text is in Norwegian, and the figure would span three pages if enlarged for readability. We may add a three-page enlargement as an annex. Fig. 3 could be enlarged for readbility, but would span two pages. The text is in Norwegian. We may add a two-page enlargement as an annex.

7. P4 Line 30 – typo halter? Should this say "halt"

Thanks for spotting this, will be changed.

8. Figure 2 – consider breaking this into 3 figures so that it can be read.

Ref. comments above.

9. P8 Line 12 – I do not think RegObs is the only open-access online real-time distribution system for avalanche forecasting (see MIN, Avanet).

As far as the authors are aware of, there is no other completely open-access online real-time distribution system. Other system lack open API's or restrict access to functionality or data partly or fully.

10. P11 Line 9 – Section D in Table 1. Typo? No section D in table 1.

Thanks, will be changed to "Table 2".

11. Figure 3 – same as comment on Figure 2 – currently the details of this figure are not readable, and they need to be since they are the basis for the testing. Break into several figures.

Ref. comments above.

12. Table 3 – I cannot find an explanation of the "4-scenario response". This needs to be clear as its not clear from just the table alone.

We will add an explanation of the two response columns in the caption. The figures are the percentage of respondents selecting the statement.

13. P12/13 – the description of how the communication effectiveness score was obtained is not clear. Despite reading several times, I remain unsure if I understand this. Ensure this method is explained well.

We will rephrase the description of the score.

14. P26 Lines 22/26 – We need to teach (1) is a poor header that does not communicate.

Thanks, we will rephrase this header.

---

## Author Comment (AC2) · 16 Aug 2018

**Open discussion** https://www.nat-hazards-earth-syst-sci-discuss.net/nhess-2018-183/#discussion

**Authors' response to RC2 review by B. Zweifel.** The response is shown in blue.

**1. Introduction**

Page 2, Line 33: Instead of Landrø et al., 2013 I would rather refer to EAWS, 2017 which would be the newer synthesis.

Thanks, we will add this reference.

Page 3, Line 2-4: Travel advises are in some regions also given within the avalanche danger assessment section of the report. There they do not resist as a general advice but more as a specific advice for the actual situation.

We will add this point in the text.

**2. Methods and data collection**

Either in this section or in the discussion at the end I encourage you to address a possible bias of your study sample towards participants with an over – average interest in avalanche safety and its possible different answers (e.g. Haegeli, Strong-Cvetich, and Haider (2012), p. 806)

We will address this point in the text.

Page 11, 12: Is there any reason how you determine the classes for positive and negative weights of the communication effectiveness score? For my feeling this was one of the weak points of the analysis. It seems a bit made up out of thin air. Maybe you can bring some more evidence how you chose this classes?

Our reasoning is that there is no objective correct answer. Accordingly, we use the expert answers, where a factor receives +1 if the majority of experts provides support, 0 for inconclusive support by the experts, and -1 if a small minority of experts regards this factor as relevant.

The rationale behind this approach was to give a penalty (a weight of -1) to statements that were selected by few/no NAWS experts and a point to statements that were selected by many NAWS experts. In the design phase, we explored using different algorithms for calculating the scores, for example by using the relative number of experts selecting the Statement as a weight. However, in order to keep the method and results relative easy to understand and interpret, we choose a straightforward approach. We will add a statement in Chapter 5, where we recommend exploring and developing better methods for quantifying the effectiveness of communication in a future study.

Page 13, Line 18, 19: I guess that this values could be biased towards a population with more avalanche expertise than average (see point described above).

We agree.

Page 12, Line 28, 29: As well here a see a possible bias towards people who use the forecast more of ten than average (see point described above).

We agree.

Page 14, Table 4: "Ski tours per year" is in my opinion only one important value of experience. It is also important to know how many years of experience someone has? Did you survey this as well? Further, I

suggest adding total values in Table 4, so that one can see how many percent of all members fell within the different competence classes.

We initially included a number of other questions and test, but had to significantly reduce the volume in order to ensure users would complete the survey. This was one of the questions we had to remove, unfortunately.

**3. Results**

I was a bit confused to find answer values from the experts here in the result section since I understood from the method section (page 11, line 4) that the expert answers just define the "correct" answers. Anyway I recognized that the comparison between the two values (from the experts and from the recreational users) is of interest of course. Therefore I propose to adopt Figure 4 and 5 in a way that you show both values (from the experts and from the recreational users) for each factor.

We have data from two surveys, the NAWS experts and the users. In Chapter 3.1 and 3,2 we present the results from the user survey, but use different names (users, recreational users, experts etc) in different ways. These inconsistencies were also pointed out by referee in RC1. We will rewrite these chapters and use

- "users" and "user respondents" instead of users, participants, recreational users and experts
- "NAWS experts" and "NAWS expert survey" instead of experts

The results from the NAWS experts will be included where relevant, but the main results are the results from the user survey. We would therefore prefer to improve the text, and keep the figures as they are.

Page 17, Line 15, 16: I assume that this point corresponds to paragraph 3.2.2 as well? Maybe you mention that here.

We will mention this.

Page 17, Line 19: I would write "Avalanche size" instead of "Size" only, so it gets clearer.

Indeed.

Page 18, Line 1-8: This is quite a bit of criticism and would demand major changes in the RegObs application. It does not really become clear, whether you intend to do so or not. I suggest you to give this point some more weight in the discussion section.

We will. The users will always ask more functionality. Several of these features will be developed in 2018.

Page 18, Figure 6: In my opinion the pictures are not really very suitable for what you want to show. The wind slab could also be a persistent slab or a new snow problem. A picture with more varying slab thickness would be clearer in my eyes. Further, the picture showing the persistent slab shows only the upper fracture. This could also easily be from a wind slab problem. You address this problem in the discussion section anyway, so I see not really a need for action here.

We agree that the pictures could be better. We recommend a separate study on the topic.

Page 19, Line 16, 17: I would prefer to have two sentences here (instead of the parentheses), it would make the text more readable.

Will do.

**4. Discussion**

Page 22, Line 15: Probably you could add a reference here which addresses the of the snow cover (e.g., Schweizer, Kronholm, Jamieson, and Birkeland, 2008).

Nice suggestions, we will do.

Page 23, Line 15: Would you have any examples of AWS who show different danger levels and/or avalanche problems and is it possible to shortly discuss pros and cons of it?

Nice suggestions, we will do.

Page 23, Line 15: Isn't there any possible conflict with red color for the core zone with the red color from danger level 4?

Could be. However, this is probably less of a problem since red is associated with danger in general and the users will quickly see that red is used in the core zone diagram independently of the danger rating.

Page 23, Line 21: A mentioned above I see the pictures as not perfectly suitable for the phenomena you want to show. This means that your chosen sample is probably not the best and I find it delicate to generalize that pictures are not suitable for this kind of communication. However, I saw that you addressed this point properly later on (line 30 ff) and accordingly do not see a necessary action here.

Ref. comment above,

Page 24, Line 27, 28: Maybe you take my comment from above with the number of years in experience into account here.

Will do.

**5. Conclusions and recommendations**

Page 26, Line 17-21 (referred to as Line 4, 5 in RC): How would you illustrate parts of regions most affected by an avalanche problem? Would you further divine the region into sub-regions? This does not become clear here.

Our study does not explore possible ways to present this information on maps, this is left as a matter for NAWS to explore. One way to do this could be to show a map of the region next to each avalanche problem where the subregion or relevant elevation interval is symbolised or colour-coded in such a manner that the user easily understand which parts of the region is most affected. If the avalanche problem is related to heavy precipitation, wind or temperature, detailed maps of these properties may show the user which parts of the region are most affected. We do use sub-regions, as a way to provide better information in the text analysis. We will probably not have information with the required detail to present higher resolution maps of danger level or avalanche problems (yet).

We will add some text on this matter.

Page 26, Line 17-21 (referred to as Line 4, 5 in RC): What do you mean with higher or lower avalanche hazard? Are you talking about another danger level or about variations within one danger level?

We are referring to another danger level, and will specify this in the text.

Page 27, Line 5, 6: I suppose here to refer to a simple "public danger scale" which has only a short characterization and a travel advice for backcountry recreationists for each danger level (e.g., https://www.slf.ch/en/avalanche-bulletin-and-snow-situation/about-the-avalanche-bulletin/danger-levels.html, https://avalanche.org/avalanche-encyclopedia/danger-scale/).

We will rephrase.

Page 27, Line 19: Concerning the communication of the core zones it would probably be worth to develop a EAWS standard design. However, I know this will be a challenge. I don't think you have to mention that in the paper anyway.

We agree.

**Technical note**

Page 3, Line 17: There is a point missing after "(DeJoy, 1999)"

Will correct.

We will include these references:

Haegeli, P., Strong-Cvetich, L., & Haider, W. (2012). How mountain snowmobilers adjust their riding preferences in response to avalanche hazard information available at different stages of backcountry trips International Snow Science Workshop 2012, Proceedings (pp. 800-806). Anchorage, AK.

Schweizer, J., Kronholm, K., Jamieson, J. B., & Birkeland, K. W. (2008). Review of spatial variability of snowpack properties and its importance for avalanche formation. Cold Regions Science and Technology, 51 (2–3), 253-272. doi:http://dx.doi.org/10.1016/j.coldregions.2007.04.009.

---

## Author Comment (AC3) · 16 Aug 2018

Open discussion https://www.nat-hazards-earth-syst-sci-discuss.net/nhess-2018-183/#discussion

Authors' response to RC3 review by M. Staples. The response is shown in blue.

**1. Introduction**

The challenges of communicating avalanche hazard to the public (section 1.2) were described very well. However, the first part of this section (lines 15-26) about impact-based warnings was confusing. If exposure and vulnerability are determined by individual users, how can AWS's issue impact-based warnings?

AWS issue regional forecasts, which describe the general probability and size of avalanches in a geographical area. These forecasts may describe the general impact for recreational activity, roads, buildings etc, but will not be able to determine the nature or scale of the impact on individuals or objects. For example, if an AWS forecasts a level 5 extreme danger in a mountain area where no one is recreating, the impact will be 0 as nobody are exposed to the danger. At the other side, the forecasts describes the impact to people that are at risk, if people choose to travel in avalanche terrain at extreme danger. So in a way the forecast is impact-based, but it cannot quantify the impact specifically as the AWS will not know how many people are exposing themselves to the risk.

We will rewrite the text to make it clearer, and possibly introduce an example.

U.S. successes (page 4, line 28) perhaps don't match other trends and could be worth mentioning. U.S system is unique in that it has very different styles and formats yet seems to be effective. The trend in the U.S. has been a declining fatality rate. The number of fatalities has been flat while use has surged, thus the rate has declined. https://avalanche.org/2016/06/27/2016627us-avalanche-fatality-trend-is-flatfor-the-past-22-seasons/

Excellent point, we will write about this and refer to the article at https://avalanche.org/2016/06/27/2016627us-avalanche-fatality-trend-is-flat-for-the-past-22-seasons/.

It would be very helpful to have an English version of Figure 2 (page 6). This would help some readers really understand the content of elements of avalanche warnings. Do any of the sections in the avalanche warning use stock language? Are they written from scratch each day? This is extremely important to know. Whether or not it contains original writing has a major effect on credibility.

We will translate the contents of the figure to English, we agree this would make it easier for the reader.

All text but that of the main message and the avalanche problems, is written manually by the NAWS forecasters. However, sometimes the forecasters may use parts of the text from the previous day, especially in the snow cover history. The text in the main message is produced in the following manner: The forecasters can get a list of text suggestions that are available depending on the chosen danger level and avalanche problem(s). He/she may then edit the text. He/she may also write the message from scratch or copy the text from the previous day. The text in the travelling advice in the avalanche problem is generated from a list of text suggestions. The selection of text is based on the combination of the chosen danger level and avalanche problem. The forecasters may edit the text afterwards. NAWS generate text suggestions in the forecast editing software for the main message and avalanche problems in order make the text in the warnings more easy for the users to read, ensure that the terms and wordings are as good and consistent as possible, make sure the time needed to produce the text is not too high and make translation of the text to English as good and easy as possible. NAWS have been having extensive discussions about to which degree should text be created from scratch by individual forecasters or rather

be predefined or suggested from a standard library of sentences and terms. NAWS is using a hybrid approach to this, and the creative from-scratch text is mostly found in the main message (Norwegian and English), avalanche danger assessment (Norwegian only) and snow pack and avalanche history (Norwegian only). However, creative text may suffer from poor language and significant individual difference that are difficult for the users to understand. Also, a lot of time may be spent writing text to convey a message that has already be written in much better way by someone else. However, the interest and motivation of users may drop if they get the feeling that too much text is auto-generated or copypaste. NAWS is very aware of this effect and continuously make efforts to avoid this from happening.

We will explain how NAWS operate and elaborate the pros and cons of stock language versus creative text.

Many issues with this paper seem to stem from a translation issue. Use of some words like "danger" and "hazard" made this paper very confusing to follow. It was sometimes unclear when the word "danger" meant specific danger rating (i.e. Low, Moderate, etc.) or a general reference to hazard. Sometimes they called it the "danger rating" other times the "danger level". While not a huge issue, this made the paper confusing.

We will revise the manuscript in order to improve this. We will make the use of "danger" and "hazard" more consistent or explain at the beginning that these terms are used interchangeably in the paper. The same applies to "forecast" and "warning" as well as "danger rating" and "danger level".

Table 1 (page 7) needs to be rewritten. This table is referenced in Figure 5 (page 16); however, the element names in Table 1 and Figure 5 do not match. They are not presented in the same order either. Both of these issues make it very difficult to understand the results of this study. This was a serious problem for me.

**Good point. We will fix this, by adding consistent names for each element in the table and the text.**

Figure 2 shows elements of the avalanche warning. Each one is labeled with a number, and the caption has a description for each element. Figure 5 shows how users ranked elements in the avalanche warning. However, it was difficult to cross-reference these two figures because the wording in the caption for Figure 2 does not exactly match the wording in Figure 5. Additionally, the elements in Figure 5 are listed in a different order than they are in Figure 2. These issues made it challenging for me to fully understand the results.

**Good point. We will fix this.**

**2. Methods and data collection**

The way the male/female demographic was described on page 9, line 10 should be changed to mirror the way it is described in page 9, line 20. Line 10 perpetuates gender biases.

**Good point. We will fix this.**

Survey design: I do not know how to design a survey, but I know the way questions are phrased can have a big effect on responses. I assume that questions on this survey were written in a neutral way. It seems very difficult to truly assess comprehension. Given this difficulty, the authors did a great job trying to quantify comprehension with their system described on page 12, lines 1-5.

**Thanks!**

Why did you test comprehension of wet slabs? Do they kill a lot of people in Norway? Persistent slab avalanches kill many people in the U.S., and they are difficult for users to understand. Risk management and travel advice messages for persistent slab avalanches are difficult to

**communicate. Additionally, there can be significant message fatigue with this avalanche type. Personally, I would have tested comprehension regarding this avalanche type.**

We wanted to test different danger levels and different avalanche problems. However, we had to reduce the scenarios to two. We agree that testing a persistent dry slab problem would be interesting. However, the wet slab problem is a relevant problem in Norway when considering high danger levels and the danger of natural avalanche.

We agree to the suggestion to make another study, in order to test more avalanche problems. It would also be useful to test more specific comprehension alternatives, and how these translate into real-life management of the avalanche threats in the field.

**Wind slabs were a good problem to test because they are so common.**

**We agree, and this is a very common problem in Norway.**

The four alternative ways to present the forecast (is it a forecast or is it an avalanche warning?) on page 11, lines 10-12 do not match items listed in Table 2, section D. For example, item 1 is listed in the table as "Avalanche danger with explanation (general advice associated with the danger level)" and it is listed on page 11 as "only the avalanche danger level and very limited travel advice". This is not a major issue, but it makes it hard for me to follow the paper. Is "general advice" the same thing as "very limited travel advice"?

**Good point. We will fix this.**

The communication effectiveness score, page 13 line 1, seemed like a great way to assess the responses from participants. Is it perfect? Who knows? Using "expert" answers as a way to evaluate participant answers seems like a great process to me.

Thanks! It is never perfect, but we thought it would be a great way to measure if the users grasped the same message as the people working for NAWS tried to convey in the warning.

**3. Results**

Another inconsistency involved the level of avalanche knowledge. The categories mentioned on page 13, lines 18-21 do not match the categories of "competence" listed in Table 4 (page 14).

**Good point. We will fix this.**

In table 4 the level of experience is categorized by "ski tours per year". How did you measure the experience of other users like snowmobilers, snow shoers, etc?

Good point. We will fix this by replacing "ski tours" with "tours in avalanche terrain". This is a translation mistake, as the question asked in the survey was independent of how the trip was carried out: "Travel in avalanche terrain - during the winter do you do 0, <5, 5-15 or more than 15 trips in avalanche terrain?".

Again, the warning elements in figure 5 do not match the elements of the warning listed in table 1 and figure 2. While the authors may be referring to the same elements, using different wording to describe the elements made it very difficult for me to understand.

**We will fix this.**

The authors did a great job summarizing the qualitative results. In the few surveys I have conducted with users in the U.S., comments often contain the most valuable info. Sometimes a single comment from a single person can be the most valuable part of the survey.

**We agree.**

Page 21, lines 9. The authors comment that "user's competence had no effect on the ranking" in line 9. Did experience have an effect? As I understand it, "competene" and "experience" are two different things.

Yes, experience is also non-significant. The F value for competence was incorrect; will be corrected to 1.966:

A user's experience had no effect on the ranking of the alternatives, F(1, 172) = .469, p = .494,  $\eta 2 = .002$ .

A user's competence had no effect on the ranking of the alternatives, F(1, 172) = 1.966, p = .163,  $\eta 2 = .010$ .

Similarly, in line 20, the authors comment that "experience did not influence the ranking." What about competence? Later in line 25 they say that "compentence had no effect on comprehension". Which is it? Is it both? This is confusing and not clear.

Also user's experience did not influence the ranking, F < 1.

We will change to: Also user's experience and competence did not influence the ranking, both F's < 1.

**4. Discussion**

Page 22, lines 20-22 – That sentence states the purpose of an avalanche warning really well.

**Thanks!**

The discussion of symbols vs text, page 23 line 25, is interesting because the tech industry has struggled with and gone back and forth on. Symbols can be confusing. Text is not, but there are issues in translating between languages.

**We agree.**

Page 24, line 15, The words danger and hazard seemed to have been used interchangeably. While they may mean the same thing, it would be better to pick one for this paper. It could be easier to use the word "danger" when referring to the danger level. This added an extra layer of confusion for me.

We will revise the manuscript in order to improve this, see comment earlier.

Page 24 lines 22-24 suggest that danger level is well understood by users. Section 3.2.3 suggests that users have difficulty understanding the danger level. Another inconsistency is that the authors would sometimes say "danger level" and other times say "danger rating". It would help to use just one.

The first point we do not consider a major discrepancy, our interpretation is that it is rather difficult to understand/respond to a level 2 warning while a level 4 is more easy to understand/respond to. We will revise the manuscript with respect to the use of "level" and "rating", see comment earlier.

All of page 25 is a great discussion. A huge issue is you addressed is when 2-3 avalanche problems are present. There are almost always 2 problems present. Great topic for further discussion in another study.

Thanks! We agree that this would be an nice topic for further studies.

**5. Conclusions and recommendations**

This study delivered some concrete findings. It seems to have been well designed, but it was challenging to understand because of the writing. The authors did a good job with their

conclusions by not presenting conclusions with too much specificity. It would be easy to read too much into the results, but they did a good job of keeping their conclusions more general.

Thanks! Your comments will be of great value for revising the manuscript.

---

## Author Comment (AC4) · 17 Aug 2018

**Open discussion** https://www.nat-hazards-earth-syst-sci-discuss.net/nhess-2018-183/#discussion

**Authors' general response.** The response is shown in blue.

**General response**

We would like to extend our gratitude to the three referees for a thorough and constructive criticism of the manuscript. The referee' comments will be used to significantly improve the manuscript in a revision.

We have addressed all points raised by the referees one by one in the comments to referees.

One point common to two of the reviews is that we should improve the explanation of the difference of the expert and user surveys. We will clarify this in chapters 2.1.1 and 2.1.2 by explaining that NAWS personnel participated in the expert survey, while all types of users participated in the user survey: Participants in the expert survey were NAWS experts (personal invitation only) and participants in the user survey were users (open invitation, anyone could participate). User survey participants included all types of users (various degree of competence and experience, from beginners/novices to experts; various types of use, from recreational to professional and preparedness). Expert survey participants included only forecasters and observers in NAWS, all trained in the same system. We will improve the description of results in Chapters 3.1 and 3.2 (and Chapters 4 and 5 where applicable), by referring to

- "users" and "user respondents" instead of users, participants, recreational users and experts
- "NAWS experts" and "NAWS expert survey" instead of experts

Any spelling errors detected will corrected, e.g. on page 26, line 18, we will remove «+» and on page 17 line 26 we will replace "Statham, 2012" with "Statham et al., 2006".

**Additional feedback**

We also received a direct feedback from Frack Techel, an avalanche forecaster and researcher at the SLF avalanche warning service, which we would like to address:

- I read with great interest your manuscript "Communicating public avalanche warnings - what works". From my perspective, particularly interesting findings were that the danger assessment (the text description!) and the avalanche problems ranked higher in importance than the danger level. This is quite different than the order shown in the EAWS information pyramid. This ranking also differs from what users knew / used in forecasts in Steiermark (Steiermark, 2015; Figure 14) or Switzerland (Winkler and Techel, 2014; Figure 5). Any idea why this seems to be different in Norway than in the Alps?

  Selection of samples could influence the results here (this is also related to the comments of RC2 under 2. Methods and data collection). The users who chose to participate in our study survey are probably above average interested in the avalanche warning. Users who check the danger level only, may be less interested in the topic and thus less likely to participate. An implication of this is that future research should explore representative samples of users of the warning in order to compare results from different user groups. Other explanations for the differences could be: (1) The Norwegian users have only five seasons (2013-2017) of experience with a public forecast and routinely use of a danger rating in Norway, while users in the Alps have decades of experience and focus on using the danger rating when discussing avalanche danger, doing avalanche training, etc. (2) The wording of the questions asked could give rise to differences. (3) The users are gradually moving from putting a major weight on the danger rating to using the

avalanche problem in the forecast, and thus not giving the rating that much importance. This may especially be the case in Norway, as the service was established at a time when the avalanche problems became popular for many services worldwide. The avalanche problems were included from the very beginning of NAWS. (4) NAWS has focused on communicating the avalanche problem and how to identify and manage the hazard, rather than the rating only. This has been natural as the mountain guides play an important role in NAWS, and use this approach in their daily work as well as their training. We will add some text discussing this in Chapter 4.1.

- I wonder whether the danger assessment being ranked so high suggests that the danger assessment is also being read frequently? Could this be related to the large percentage of experienced and professional survey respondents?

    This could be an explanation. Another possibility is that many users read the assessment in order to learn more about avalanche danger, which factors are important and what causes the danger and changes in danger (this is based on feedback from several user surveys we have done previously).

- You state that these rankings also persist when you stratify by user experience This is somewhat in contrast to Hallandvik et al., 2017 (Table 3) who showed that novices ranked danger level more than avalanche problems, and experts vice versa. Maybe you could comment on this when revising the manuscript.

    We agree, there is a difference in our results. The Hallandvik study was based on "an online survey conducted during an ad-hoc avalanche seminar in Sogndal on January 31, 2015, four days after a significant avalanche cycle with several naturally and human triggered avalanches occurred in the area. Sogndal is a popular area for backcountry and freeride skiing in Western Norway". Our study is based on an open invite, not targeting a specific group of people at one geographical location. The Hallandvik study was conducted after 2.2 seasons of public forecasting in Norway, while our study was conducted after 5 seasons of public forecasting. These factors may affect the sample of respondents available for the surveys, and the results. We will add some text discussing this in Chapter 4.1.

- Considering the expert respondents, the avalanche problem was considered by a very large proportion as important (79%). What was the importance frequency of the danger level in this group?

    It was 25 %. We will add results from NAWS expert survey for comparison.

The authors.

---

## Author Response (AR1)

**Open discussion** https://www.nat-hazards-earth-syst-sci-discuss.net/nhess-2018-183/#discussion

**Authors' response to revised manuscript.**

The manuscript has been revised according the comments and suggestions received during the open discussion, c.f. the list at
5   the end of this document. All comments are addressed. Additionally, identified spelling mistakes have been corrected and unclear language has been improved. Figures have been improved. During the revision, a missing data filter was identified in the section 3.2.1 and 3.2.2 analysis. This has been corrected. The change in results was minor and did not affect the discussion or conclusions, but the percentages and sample size reported for "users" have been updated in section 3.2.1 and 3.2.2, conclusion and abstract.

10   The revised manuscript is provided in word format, including the text, the seven figures and the five tables. We also provide a

The authors have the following ORCID, we would appreciate if this could be included in the author list:

-   Rune Engeset: https://orcid.org/0000-0003-2608-2895

-   Gerit Pfuhl: https://orcid.org/0000-0002-3271-6447

15   -   Audun Hetland: https://orcid.org/0000-0003-1299-8077

-   Andrea Mannberg: https://orcid.org/0000-0003-3703-7416

We hope the revised text and figures are according to the NHESS regulations.

**List of point-by-point response to the reviews, and list of all relevant changes:**

General comments:

20   1.   General response: We have explained and systematically described the two groups of respondents as "users" and "NAWS experts" in order to clarify. We have also improved the description of the survey.

2.   Additional feedback: We have added text addressing these comments in section 4.1 and added NAWS expert results to figures 4 and 5.

RC1:

25   1.   The use of expert survey: We have improved the text and the description of the score. We have added text and a separate section 2.4 on this. Section 2.1 has been rewritten (cf. comment 1 above)

2.    This point is addressed by adding section 2.6 and adding recommendation to chapter 5 in section 2.1 and has been improved.

3.   Additional: All comments have been addressed.

RC2:

1. Intro: All points have been addressed.

2. Methods: All points have been addressed.

3. Results: All points have been addressed.

4. Discussion: All points have been addressed.

5. Conclusions: All points have been addressed.

6. Technical note: All points have been addressed.

RC3:

1. Intro: All points have been addressed.

2. Methods: All points have been addressed.

3. Results: All points have been addressed.

4. Discussion: All points have been addressed.

5. Conclusions: All points have been addressed.

The authors.

**Communicating public avalanche warnings – what works?**

[revised manuscript text omitted]

**1.2 Warning and risk communication**

The purpose of warnings is to inform people at risk about the hazard and to promote "correct", and safe behavior (Wogalter et al., 2005). To do so warnings may not only assess threat and danger, but also exposure and vulnerability (WMO, 2015). Such impact-based warnings have been shown to be more effective than other types of warnings and are more and more in demand (DeJoy, 1999). Impact-based warnings facilitates informed decision-making, which in turn leads to desirable outcomes and prevents unnecessary costs to society (Pielke and Carbone, 2002). In the case of avalanche warnings they provide users with both general and specific information about the current and expected level of avalanche danger, the type of avalanche problem at hand, and with behavioral advice. The main aim of the warning message is to inform the user about the nature and severity of current and expected threats, and about how he or she can mitigate the risk or avoid the threats. However, since most regional AWS' do not provide specific and local descriptions of the forecasted risk, it may be difficult to effectively reach this goal. In addition, most AWS' lack detailed information on the type and number of individuals who are at risk, and on the exposure and vulnerability of these. Thus, most AWS' provide impact-based warnings in a general sense, but not in terms of impact specific to detailed geographical locations, people, roads and so on. AWS issue regional forecasts, which describe the general probability and size of avalanches in a geographical area. These forecasts may describe the general impact for recreational activity, roads, buildings etc., but will not be able to determine the nature or scale of the impact on individuals or objects. For example, if an AWS issues a warning at level 5 extreme danger in a mountain area where no one is recreating, the impact will be nil as nobody is exposed to the danger. At the other side, this warning describes the impact to people that are at risk, if people choose to travel in avalanche terrain at extreme danger. So in a way the forecast is impact-based, but it cannot quantify
* * *
**Flyttet ned [1]:** Impact-based hydro-meteorological hazard warnings are getting more and more in demand, as are impact-based warnings for other geohazards. Impact-based warnings do not only assess threat and danger, but also exposure and vulnerability (WMO, 2015), and have been shown to be more effective than other types of warnings (DeJoy, 1999).

**Flyttet (innsetting) [1]**

**Slettet:** Impact-based hydro-meteorological hazard warnings are getting more and more in demand, as are impact-based warnings for other geohazards. Impact-based warnings do not only assess threat and danger, but also exposure and vulnerability (WMO, 2015), and have been shown to be more effective than other types of warnings (DeJoy, 1999). Informed decision-making leads to desirable outcomes and prevents unnecessary costs to society (Pielke and Carbone, 2002). Avalanche warnings provide

**Slettet:** ¶

the impact specifically as the AWS will not know how many people are exposing themselves to the risk. The warnings advice the users on how to reduce or avoid being exposed and vulnerable to the avalanche danger, and thus the risk.

[revised manuscript text omitted]

In other states, such as in the U.S. and several European countries, Avalanche Warnings Services (AWS') have succeeded to avoid an increase in fatalities, although their warning styles and formats have varied quite a bit. The trend in the U.S. has been a declining fatality rate: whereas the number of fatalities has been rather constant, the use of avalanche terrain has surged (Birkeland, 2016).

In order to halt the undesirable trend in avalanche accidents in Norway, the Norwegian Government in a white paper in 2012 decided to establish the Norwegian Avalanche Warnings Service (NAWS) in January 2013 (Engeset, 2013). NAWS publishes regional avalanche warnings for Norway, including Svalbard, on a daily basis on the web portal www.varsom.no (Johnsen, 2013). The Norwegian Water Resources and Energy Directorate owns and operates NAWS, in collaboration with the Norwegian Public Roads Authorities and the Norwegian Meteorological Institute. The reduction in annual fatalities during the previous 3-4 years suggest that NAWS is effective, as the accidents numbers have not increased although the use of avalanche terrain for ski touring has increased drastically in Norway, every year during the last decade years or so.

In 2017, regional avalanche warnings were issued for 21 regions in Norway (Fig. 1). In addition, warnings were issued for the rest of the country when the avalanche danger was expected to reach danger level 4 or 5. An example of an avalanche warning on Varsom.no is shown in Fig. 2. The avalanche warning published on Varsom.no includes the elements described in Fig. 2 and Table 1.

Slettet: .no

Slettet: .

Slettet: ¶

Slettet: er

Slettet: was established

[Figure]

**Figure 1.** Screen dump from Varsom.no showing the avalanche warning regions in Norway. Daily warnings are published for the regions coloured according to the danger level. The other regions (grey) are monitored and warnings are only published at danger levels 4 and 5.

**Formatert:** Bildetekst

**Slettet:** An example of

**Slettet:** for

**Slettet:** (screen dump from Varsom.no)

**Slettet:** ¶

**Slettet:**

[Figure]

**Formatert:** Venstre

[revised manuscript text omitted]

Formatert: Skrift: Fet
Slettet: flash, the
Slettet: The a
Formatert: Skrift: Fet
Slettet: ,
Slettet: which is a
Formatert: Skrift: Fet
Formatert: Skrift: Fet
Formatert: Skrift: Fet
Slettet: The s
Formatert: Skrift: Fet
Slettet: s
Slettet: and recent avalanche
Formatert: Skrift: Fet
Slettet: .
Slettet: The m
Formatert: Skrift: Fet
Slettet: .
Formatert: Skrift: Fet

quality. However, to date, no formal evaluation has been done of how effective NAWS is at communicating its intended message. Such an evaluation is important, as public avalanche warnings have only been available in Norway since 2013 and Norwegian users are likely less used to using the warnings to manage risk than users in countries with a longer history of public avalanche warnings.

[revised manuscript text omitted]

**(a)**

**I**

**Level 2 Moderate avalanche danger**

Travel in avalanche terrain requires knowledge, experience in route finding and ability to identify the avalanche problem. Generally we recommend to avoid terrain steeper than 30 degrees in order to not release an avalanche.

**Wind slab**

Buried weak layer of new snow

[Figure]

Avalanche type — Slab avalanches
Avalanche size — 2- Small
Triggering — Low additional load
Distribution — Some steep slopes
Probability — Possible

Be careful in areas steeper than 30 degrees with wind slabs until the snow is stabilised. Keep distance between each other when travelling in steep terrain. The likelihood of releasing an avalanche is highest in convex terrain and where the wind slabs are soft. Look out for areas where wind recently has deposited snow, typically behind ridges, ribs and gullies. Local wind effects and changing wind direction can give large variation in where the wind slabs are formed. Snow cracking up around the skis/board is an obvious clue.

**Wind slab**

Buried weak layer of new snow

[Figure]

Avalanche type — Slab avalanches
Avalanche size — 2- Small
Triggering — Low additional load
Distribution — Some steep slopes
Probability — Possible

**Wind slab**

Be careful in areas steeper than 30 degrees with wind slabs until the snow is stabilised. Keep distance between each other when travelling in steep terrain. The likelihood of releasing an avalanche is highest in convex terrain and where the wind slabs are soft. Look out for areas where wind recently has deposited snow, typically behind ridges, ribs and gullies. Local wind effects and changing wind direction can give large variation in where the wind slabs are formed. Snow cracking up around the skis/board is an obvious clue.

**(b)**

**I**

**Level 4 High avalanche danger**

Travel in avalanche terrain is not recommended. Avalanches may reach avalanche-prone roads and buildings. Avalanche-prone roads may be closed and houses may be evacuated. Electricity supply and communication may be affected.

**Wet slab**

Water pooling in/above snow layers

[Figure]

Avalanche type — Slab avalanches
Avalanche size — 3- Medium
Triggering — Spontaneous release
Distribution — Many steep slopes
Probability — Likely

Keep away from avalanche terrain (terrain steeper than 30 degrees) and avalanche runout zones. Infrastructure: Large natural avalanches may occur and reach roads and houses.

**Wet slab**

Water pooling in/above snow layers

[Figure]

Avalanche type — Slab avalanches
Avalanche size — 3- Medium
Triggering — Spontaneous release
Distribution — Many steep slopes
Probability — Likely

**Wet slab**

Keep away from avalanche terrain (terrain steeper than 30 degrees) and avalanche runout zones. Infrastructure: Large natural avalanches may occur and reach roads and houses.

[Figure]

**Slettet:** ¶
*<objekt>*

**I**

Region Troms: Faregrad 2 - moderat skredfare

Ferdsel i skredterreng krever kunnskap, erfaring i rutevalg og evne til å identifisere skredproblem. Generelt anbefales det å unngå terreng brattere enn 30 grader for å unngå å løse ut skred.

Fokksnø

Nedføyket svakt lag med nysnø

Skredtype: Flakskred
Skredstørrelse: 2 - Små
Utløsningsårsak: Liten tilleggsbelastning
Utbredelse: Noen bratte heng
Sannsynlighet: Mulig

Vær forsiktig i områder brattere enn 30 grader med fokksnø til den har fått stabilisert seg. Hold avstand mellom hverandre ved ferdsel i bratt terreng. Det er størst sannsynlighet for å løse ut skred på kul-formasjoner i terrenget og der fokksnøen er myk. Se etter områder hvor vinden nylig har lagt fra seg fokksnø, typisk bak rygger, i renneformasjoner og søkk. Lokale vindeffekter og skiftende vindretning kan gi stor variasjon i hvor fokksnøen legger seg. Snø som sprekker opp rundt skiene/brettet er et typisk tegn.

Fokksnø

Nedføyket svakt lag med nysnø

Skredtype: Flakskred
Skredstørrelse: 2 - Små
Utløsningsårsak: Liten tillegg
Utbredelse: Noen bratte
Sannsynlighet: Mulig

Fokksnø

Vær forsiktig i områder brattere enn 30 grader med fokksnø til den har fått stabilisert seg. Hold avstand mellom hverandre ved ferdsel i bratt terreng. Det er størst sannsynlighet for å løse ut skred på formasjoner i terrenget og der fokksnøen er myk. Se etter områder hvor vinden nylig har lagt fra seg fokksnø, typisk bak rygger, i renneformasjoner og søkk. Lokale vindeffekter og skiftende vindretning kan gi stor variasjon i hvor fokksnøen legger seg. Snø som sprekker opp rundt skiene/brett er et typisk tegn.

**I**

Region Troms: Faregrad 4 - stor skredfare

Ferdsel i skredterreng anbefales ikke. Skred kan treffe skredutsatte veier og bebyggelse. Flere skredutsatte veier ventes stengt og bebyggelse kan bli evakuert. Strømforsyning og kommunikasjon kan bli rammet.

Våte flakskred

Opphopning av vann i/over lag i snødekket

Skredtype: Flakskred
Skredstørrelse: 3 - Middels
Utløsningsårsak: Naturlig utløst
Utbredelse: Mange bratte heng
Sannsynlighet: Sannsynlig

Våte flakskred

Opphopning av vann i/over lag i snødekket

Skredtype: Flakskred
Skredstørrelse: 3 - Middels
Utløsningsårsak: Naturlig utløst
Utbredelse: Mange bratte
Sannsynlighet: Sannsynlig

Våte flakskred

Hold deg unna skredterreng (brattere enn 30 grader) og utløpssoner for skred. For infrastruktur: naturlig utløste skred kan forekomme og nå ned til vei/bebyggelse.

Hold deg unna skredterreng (brattere enn 30 grader) og utløpssoner for skred. For infrastruktur: Store naturlig utløste skred kan forekomme og nå ned til vei/bebyggelse.

**Figure 3.** Alternatives 1 to 4 used for the two scenarios: (a) level 2 and wind slab (upper panel) and (b) level 4 and wet slabs (lower panel). The text has been translated from Norwegian to English.

After the expert respondent had read the example, we first asked him or her to rate how well the danger was communicated in the example, on a scale from 0 to 10. We thereafter asked the expert to identify key information elements and behavioural implications of the avalanche forecast. The options were predefined, as described in Table 3. We were specifically interested in identifying the most important message that the forecast aimed to communicate.

**2.4 Communication effectiveness score**

In order to establish a communication effectiveness score, we used the NAWS expert answers to allocate weights to the different behavioural implications. We allocated positive weight of +1 to elements positively identified as important by more than one out of three experts (33 %), and a weight of -1 to elements positively identified by less than one out of 5 experts (20 %). All other elements were given a weight of null.

Our reasoning behind using positive and negative weights to calculate the communication effectiveness score, is that there is no objective correct answer. Accordingly, we use the NAWS expert answers, where a factor receives +1 if the majority of experts provides support, 0 for inconclusive support by the experts, and -1 if a small minority of experts regards this factor as relevant. The rationale behind this approach was to give a penalty (a weight of -1) to statements that were selected by few/no NAWS experts and a point to statements that were selected by many NAWS experts. In the design phase, we explored using different algorithms for calculating the scores, for example by using the relative number of experts selecting the statement as a weight or decimal weights. However, in order to keep the method and results relative easy to understand and interpret, we choose a straightforward approach.

The expert choices and resulting weights are listed in Table 3. As can be seen in Table 3, many experts agreed on the most important implications, and very few items are therefore close to the cut-off value. Nevertheless, to ensure that our results do not hinge on our chosen levels (33% and 20%), we have tested both upward and downward variations of the cut-off values. The results presented in section 3 are robust to these variations.

**Table 3.** Expert survey results (number of respondents selecting the statement, in percent) and design weights established for a communication effectiveness score.

| Statement | 2-scenario response | 2-scenario weight | 4-scenario response | 4-scenario weight |
| --- | --- | --- | --- | --- |
| 1. Unngå alle løsneområder (*Avoid all release areas*) | 20 % | -1 | 84 % | +1 |
| 2. Unngå noen løsneområder (*Avoid some release areas*) | 63 % | +1 | 9 % | -1 |
| 3. Unngå alle utløpsområder (*Avoid all runout areas*) | 8 % | -1 | 84 % | +1 |
| 4. Unngå noen utløpsområder (*Avoid some runout areas*) | 39 % | +1 | 11 % | -1 |
| 5. Unngå skredutsatte veier (Avoid avalanche-exposed roads) | 6 % | -1 | 75 % | +1 |

**Slettet:** left

**Slettet:** ¶

**Slettet:** table

| | | | | |
|---|---|---|---|---|
| 6. Kunne mye om snø for å vite hva jeg skal unngå (*Know a lot about snow in order to know what to avoid*) | 29 % | 0 | 16 % | -1 |
| 7. Grave i snøen for å vite hva jeg skal unngå (*Dig in the snow in order to know what to avoid*) | 12 % | -1 | 6 % | -1 |
| 8. Vite mye om været siste to dager for å velge terreng (*Know a lot about the weather the last two days in order to choose terrain*) | 45 % | +1 | 13 % | -1 |
| 9. Forvente store lokale forskjeller (*Expect large local variability*) | 71 % | +1 | 16 % | -1 |

**2.5 User survey**

The user survey was open to the public during the period 1 November – 15 December 2017. We published links to the survey on a relatively wide set of platforms: Varsom.no, the free online skiing magazine friflyt.no, and on the Facebook page of the most popular weather service in Norway, YR.no. The association of snow scooter clubs (Skuterklubbenes fellesråd) and the Norwegian Hiking Association (DNT) kindly distributed the survey to their members. Finally, we announced the survey on the Nordic avalanche conference in Åndalsnes in the beginning of November.

Each participant was asked to answer the full survey (section A-E). In section D, the users were, just like the experts, randomly exposed to one out of four alternative ways of presenting the avalanche forecast for the level 2 and level 4 scenarios, and thereafter to first rank how well the danger was communicated on a scale from 1 to 10, and to mark the most important behavioural implications of the forecast.

We used the weights in Table 3 to calculate a "communication effectiveness score" for each participant, and each behavioural implication. To illustrate, consider a user respondent who ticked the boxes for statements 1, 2 and 3 after reading an example of the level 2-scenario. Based on the scores in Table 3, we would give this user a score of -1 (the sum of -1+1-1). If the user instead ticked the boxes for statements 3 and 5 after reading an example of the level 4-scenario, we would give him or her a score of +2 (the sum of +1+1). The scores for the level 2-scenario ranged from -4 to +4, and for the level 4-scenario from -6 to +3.

**2.6 Web survey or field testing**

Our over aching aim for this study was to investigate users comprehension of the warning. Ultimately, all public warnings aim at making people take the correct actions at the correct time. However, there is a large body of evidence demonstrating that there is a mismatch between what people say and what they do (e.g. Jerolmack and Khan, 2014). Therefore, if we studied people's behaviour and not comprehension we would not know if the lack of correct action was due to lack of compression or rather a mismatch between attitudes and behaviour.

Self-reports is by many accounts not a perfect method, but in this case we found it to be the best approach to test peoples comprehension. In addition it allows us to collect a substantially larger number of respondents compared to for example a

field-study or interviews. A web-based survey is also relevant, as many decisions are made based on the avalanche warning ahead of getting into navigation in the terrain. It could be decisions such as to choose forest rather than the alpine for today's trip, or delay the planned trip a few days until the snow stabilizes. However, it would be very interesting to test what people know and also what they do. This would call for a different study all together, but is a very good idea for future research.

**2.7 Ethics**

This study registered anonymous information exclusively and did not collect data that can be used to identify individuals. All respondents actively gave their consent for the use of the data for research and the project.

**3. Results**

In this chapter, we present the avalanche-related demographics of the user respondents (Section A and E), well-functioning and malfunctioning parts of the 2017-version of the avalanche warnings on Varsom.no, as perceived by the participants (Section B), the participants' evaluation of how well text, symbols and pictures assist the informational content in the warnings (data from Section C), the participants' evaluations of  how well different levels of complexity in the text persuade the informational content in the warnings  (data from Section D), and test results for level of comprehension at different levels of complexity in the warning texts (also data from Section D).

**3.1 Demographics**

The statistics of the users with respect to competence, experience, activities and geography are listed below and in Table 4:

- 14 % of the user respondents had no or little avalanche knowledge (labelled "None" in Table 4), 27 % stated that they had avalanche related competence but no formal training ("Competent, no course"), 48 % stated that they had avalanche related competence and formal training ("Competent, course"), and 10 % were avalanche instructors or professionals ("Expert").
- 82 % stated that they had used avalanche gear (e.g., avalanche beacon, shovel and probe) for several seasons, 7 % one season only, and 11 % had never used this type of equipment.
- The majority of users stated that their main activity in avalanche terrain was alpine ski touring (66 %). Relatively many users also stated that they engaged in off-piste skiing (32 %), or Nordic mountain skiing (23 %), while relatively few said that they travel in avalanche terrain by foot (9 %), on a snowmobile (7 %), or on snow shoes (3 %). Three percent stated that they engage in other types of activities in avalanche terrain. Note that the users could chose multiple activities.
- Concerning the use of NAWS, 76 % of the users answered that they always use the avalanche warnings, 21 % use the warnings on a regular basis, and 3 % answered that they rarely read the forecast.

**Slettet:** Testing the understanding of an avalanche warning by analysing actions and decisions in the field would be interesting. However, we aimed at testing the efficiency of the warnings in generally, for which a web-based survey is appropriate as it is less complex and easier to eliminate factors that are not attributable to the communication of the warning. Each method has its advantages and disadvantages. Web-based surveys are cost and time efficient, show high response rates and ensure voluntarily responding as the user without any costs can abort the study. Importantly, anonymous web surveys compared to active tracking in the field has high data security, i.e. we do not need to ask people to carry recording devices on their trip or requesting data logs from apps, which allows identifying the user. This often leads to more honest answers, as users are aware that they are anonymous, which is not the case in fieldwork. Field-based testing also has other challenges. There is a large body of evidence showing that there is a major difference between behavioural intentions/attitudes and actual behaviour. Therefore, if we would focus solely on the behaviour in the field and found a miss-match between this and the communicated warning, we may not know where along the line from information to action things went wrong - if this was due to lack of comprehension or rather risk seeking attitude. Furthermore, a web-based survey is relevant, as many decisions are made based on the avalanche warning ahead of getting into navigation in the terrain. It could be decisions such as to choose forest rather than the alpine for today's trip, or delay the planned trip a few days until the snow stabilises.¶

**Slettet:** 5

**Slettet:** recreational

**Slettet:** some

**Slettet:** participants

**Slettet:** respondents

**Slettet:** respondents

**Slettet:** recreational

Many of the respondents have avalanche related competence and formal training, and many use the avalanche warnings on a regular basis. This suggests that the sample of respondents could be biased towards a population with more avalanche expertise than average.

5  **Table 4.** Contingency table of user respondents' experience (number of tours in avalanche terrain per year) versus competence.

| Competence | Experience (tours in avalanche terrain per year) | | | | |
|---|---|---|---|---|---|
| | 0 | < 5 | 5-15 | > 15 | N |
| None | 10% | 48% | 35% | 8% | 40 |
| Competent, no course | 6% | 23% | 41% | 30% | 81 |
| Competent, course | 0% | 6% | 39% | 55% | 121 |
| Expert | 0% | 5% | 14% | 82% | 22 |

**3.2 Avalanche warning**

A total of 264 user respondents completed the questions in Section B. In this section, we asked the respondents to identify risk factors that they perceived difficult to manage or mitigate, parts of the avalanche warnings that they perceived difficult to
10  understand, and important information perceived to be missing in the avalanche warnings. Key results from the NAWS expert survey are also presented in this chapter for comparison.

**3.2.1 Avalanche risk factors considered difficult to assess and manage**

In order to find out what the users consider as being most difficult to assess and manage, we asked "Which factors are most difficult to assess and manage in order to complete a safe trip?". The respondents could choose multiple factors. Available factors and results are shown in Fig. 4. The results show that

- The vast majority (87 %) of the users perceives that the *snow cover* is the single most difficult factor to assess and manage. This judgement does not depend on the respondent's experience or competence (Chi-square test, p = .516 and p = .403, respectively). 86 % of the NAWS experts considered this factor as the most difficult factor.
- 34 % of the users perceive that *other people in the group* is the most problematic factor. More than every second NAWS expert (51 %) rated this as the most difficult.
- Among the users, there is a relatively even distribution of individuals who perceive that *terrain traps* (28 %), and *weather* (25 %) constitute the other most problematic factors.
- Steepness is perceived as a problematic factor among relatively few respondents.

[Figure]

**Figure 4.** Factors users and NAWS experts considered difficult to assess and manage in order to have a safe trip in avalanche terrain.

**Slettet:** 79

**Slettet:** both recreational participants and experts (87 % of the experts and 79 % of the recreational users)

**Slettet:** 7

**Slettet:** ,

**Slettet:** 2

**Slettet:** More than every second expert (52 %)

**Slettet:** 2

**Slettet:** , while 32 % of the recreational users

**Slettet:** recreational

**Slettet:** 26

**Slettet:** 21

[Figure]

**Slettet:**

**Slettet:** recreational

**Slettet:** users

**3.2.2 Avalanche risk factors considered most and least important**

In order to find out what the users consider as being the *most important element in the warning*, we asked "Which elements in the avalanche warning are most important?". The respondents could choose multiple answers. Alternatives and results are presented in Fig. 5. The results show that the users perceive a relatively wide range of elements in the warning to be important.

- A majority of the users state that the *avalanche assessment* (69 %), the *avalanche problems* (67 %) and the *main message* (65 %) constitute the three most important elements in the warning,
- About half of the users consider the *snow cover history* (56 %) and the *danger level* (48 %) as important.
- Over a third of the users consider *snow and avalanche observations* (37 %), *mountain weather* (39 %) and *management advice* (42 %) as important.

10 We find no evidence that the elements chosen as most important depended on age, gender or experience (linear regression $R^2$ = .022, p = .224). The NAWS experts rated the *avalanche problems* as the most important factor (77 %), followed by the *avalanche assessment* (62 %), the *main message* (57 %) and the *snow cover history* (55 %). The *danger level* was considered most important by 39 % only.

[Figure]

**Figure 5.** Factors the users and the NAWS experts considered most important in the avalanche warning on Varsom.no.

**Slettet:** participants

**Slettet:** recreational participants

**Slettet:** 68

**Slettet:** 65

**Slettet:** 2

**Slettet:** recreational respondents

**Slettet:** *and avalanche analysis*

**Slettet:** recreational respondents

**Slettet:** 40

**Slettet:** 41

**Slettet:** The results from t

**Formatert:** Skrift: Ikke Kursiv

**Slettet:** respondents were similar, with

**Formatert:** Skrift: Ikke Kursiv

**Slettet:** rated by far

**Slettet:** 9

**Formatert:** Skrift: Kursiv

**Slettet:** *and avalanche analysis*

**Formatert:** Skrift: Kursiv

**Slettet:** 25

**Slettet:** We find no evidence that the elements chosen as most important depended on age, gender or experience (linear regression $R^2$ = .022, p = .224).

**Slettet:** ¶

[Figure]

**Slettet:**

In order to find out what the users consider of *least importance or use*, we asked "Was anything of little use or importance? You may elaborate on the problem being format, content or other." A total of 69 participants responded to this question. 20 of these provided positive or neutral comments. We summarize the critical feedback, and our interpretation of this feedback, below.

- Seven users stated that they found *the mountain weather to be superfluous*, and that they rather used the standard weather forecast. Thus, clarification in the difference between the weather forecast and the summary of the mountain weather, and the link between the mountain weather history and forecast, and the avalanche forecast, is recommended.

- Five users stated that the warning contained *too many, and complex details and information*. These users were mainly novices. This may imply that users with less skills and interest in the topic fail to get the key messages.

- However, another set of six users considered the level of detail as *too low*. These users stated that the usefulness of the warning would be higher if it were less general, and if the forecast region was smaller. These answers point to the possibility that general forecasts for relatively large regions reduce the attention paid to the warnings.

- Three users found the *core zone sector diagram* to be problematic. More specifically, these users found it difficult to know if dark sectors represent safe, or unsafe regions. Although only three users commented on this, their feedback is important since it implies that some users of NAWS may chose the unsafe sector because they misunderstand the graphics. See also section 3.2.3 for related results.

- Finally, four users found the *snow and avalanche observations* sometimes be *too complicated* or described in *too difficult terms*.

**3.2.3 Elements easily misunderstood or poorly communicated**

A total of 95 users provided comments on if the avalanche warning contains parts that are *easily misunderstood or poorly communicated*. 30 of the comments were positive or neutral. We summarize the critical feedback, and our interpretation of the comments, below.

- Eleven users found the *core zone sector diagram* to be easily misunderstood. Like in the case of users who stated that the core sector diagram to be of little use, these users stated that they found it difficult to know which of the sectors (dark or light) that are most dangerous. Some users suggested to add a legend or use more or different colours. These findings corroborate the findings in 3.2.2.

- Another 11 users perceived that the regional warnings provide *too little details in terms of spatial or temporal variability*, and that the *forecasted regions were too large*. These findings corroborates the findings in 3.2.2.

- Eight users found it difficult to understand the *danger level*, in terms of the meaning and consequence of it for the user. This is important, because if users do not understand the meaning of the danger level, they are poorly equipped to manage their risk exposure.

**Slettet:** respondents

**Slettet:** respondents
**Slettet:** participants

**Slettet:** respondents
**Slettet:** participants

**Slettet:** participants
**Slettet:** participants
**Slettet:** participants

**Slettet:** respondents

**Slettet:** participants

**Slettet:** 11
**Slettet:** recreational respondents
**Slettet:** participants
**Slettet:** participants
**Slettet:** participants
**Slettet:** recreational respondents

**Slettet:** recreational respondents

- Finally, six users stated that the large amount of information provided in the warning made it difficult, especially for beginners, to decipher the key message. This corroborates the findings reported in section 3.2.2, where five users stated that the warning contained too much details and information.

The answers from the NAWS expert survey suggest that experts perceive similar factors to be problematic as users do: i.e., the *core sector and elevation diagrams*, *spatial and temporal variability*, the *danger level*, and *uncertainty*). However, the NAWS experts also pointed to a few problematic factors not mentioned by the users: *Avalanche size* (especially the name "small" used for size 2), *probability* and *distribution*. Note that the EAWS is changing the denominations used for avalanche sizes during 2018, which will resolve the problem with communicating size 2 avalanches.

**3.2.4 Missing information and features**

In the final part of section B, we asked the respondents to identify missing information in the avalanche warning. 67 respondents provided comments. About 20 of these stated that no important information was missing. The elements asked for by the remaining 47 participants were the following:

- Observed weather and snow, and links to more detailed observations
- ATES recommendations (Avalanche Terrain Exposure Scale is a method for classifying the degree of terrain avalanche-exposure, Statham et al., 2006)
- Advice connected to competence levels, and
- More detailed warnings/information. Better visualisation of important weak layers (depth, type, etc)

We also asked the participants if some information or features are missing in the RegObs application. 81 users responded to this question, of which about 35 responded they did not use the application or were indifferent. The users asked for the following to be included in future releases:

- Weather data,
- A possibility to enter and record snow profiles,
- A possibility to read the avalanche warning (at least the danger level and avalanche problems) in the application,
- An opportunity to track trips,
- A more user-friendly interface,
- Access to avalanches and avalanche paths,
- Information about actual elevation in relation to the avalanche problem elevation range, and
- Easy access to the snow cover history and relevant recent snow profiles nearby.

Several of these features are being implemented by the time of publication of this study.

The results from the NAWS expert survey suggested that these pieces of information in demand:

- More precise description of where in the region or terrain the avalanche problem is expected, and where the danger level is expected to be lower, and

Slettet: recreational

Slettet: (

Slettet: recreational

Slettet: recreational

Slettet: S

Slettet: i

Slettet: 2012

Slettet: recreational

Slettet: found the

Slettet: be missing

Formatert: Skrift: Ikke Kursiv

Formatert: Skrift: Ikke Kursiv

Slettet: are missing

- A better description of the uncertainty and local variability.

**3.3 Testing of text versus symbols or pictures**

In section C, we asked the respondents to rate how well text, icons, and pictures communicate the avalanche problem on a scale from 1= poor, to 3 = good. Each respondent evaluated two types of avalanche problems: a wind slab, and a persistent slab (see Fig. 6). A total of 222 user respondents completed this section of the survey.

[Figure]

| | (a) | (b) | (c) |

**Figure 6.** Test of what communicates the avalanche problem best in the avalanche warning: (a) text, (b) symbols or (c) pictures.

The results show that users preferred text and symbols to pictures (Table 5). 89 % rated the new EAWS symbols as good or OK. The users were familiar with the names of the avalanche problems, which have been presented as text on Varsom.no during the previous three seasons. The users were not familiar with the symbols, as they were introduced at Varsom.no for the 2017/2018-season after becoming introduced as an EAWS standard in June 2017 (EAWS, 2017b). Pictures have not been used in the warning on Varsom.no, but a few users may have seen the pictures on the avalanche school at Varsom.no. Notably, we found that the symbols were rated more positively the more experienced a respondent was, $\chi^2_{203}$ = 15.26, p = .018. The text and pictures were rated equally irrespective of one's experience, p =.338 and p = .543, respectively.

**Table 5.** Results from test of what communicates the avalanche problem best in the avalanche warning.

| Text | Symbol | Picture | Rating |
|------|--------|---------|--------|
| 46 % | 51 % | 38 % | Good |
| 40 % | 38 % | 36 % | OK |
| 14 % | 11 % | 25 % | Poor |

**Slettet:** recreational

**Slettet:** 6

**Slettet:** less

**Slettet:** 6

**3.4 Testing of comprehension of the two scenarios**

A total of 177 user respondents completed the test for comprehension in Section D, by responding to one of the four alternatives for each of the two scenarios. To recap, we asked the respondents to 1) rate how well they perceived that the avalanche danger was communicated, 2) what the most important behavioral implications of the warning was, and 3) what advise they would give to others based on the warning message. We measured how well the forecast persuaded the warning on a scale from 1 to 10. Of those who provided answers to this question, 21 % gave a rating of 10, and 56 % a rating of 8 or higher. Only 14 % gave a rating of 4 or lower. Mean ratings for the two scenarios (danger level 2, and level 4), and for each of the four alternatives are presented in Fig. 7 below (left column). Fig. 7 also depicts the comprehension scores (right column). Higher scores indicate a higher match between the behavioral implications chosen by the users and the NAWS experts. For the danger level 2-scenario the minimum score is -4 and the maximum score is + 4, while scores for the danger level 4-scenario range from -6 to +3.

[revised manuscript text omitted]

3. *Local information matters.* Problem: The avalanche warnings are produced for relatively large geographical areas with big spatial variations in the snow cover. Possible solution: Use maps to show the parts of the region (subregions or elevation intervals) that are most affected by the avalanche problem(s), or where the avalanche danger is expected to be one value higher or lower than the rest of the region. Maps could show which parts are most affected, by showing the properties creating the avalanche problems, e.g. heavy precipitation, wind, temperature, etc. NAWS could use sub-regions, as a way to provide better information in the text analysis. NAWS will probably not have information with the required detail to present higher resolution maps of danger level or avalanche problems yet. Another way could be to present local weather history, and/or snow observations from automatic stations, or to present the snow history by visualising some manual snow observations as time series.

4. *We need to teach snow dynamics.* Problem: A very large share of respondents state that they find it most difficult to assess and manage the snow cover. Possible solution: Present the avalanche problem, snow cover analysis and the avalanche danger assessment in a more systematic and pedagogical manner in order to improve the competence of the users. It should be noted that even the NAWS experts considered the snow cover as the most difficult factor, suggesting that it is complex to manage for users at all levels.

**Slettet:** recreational

**Slettet:** recreational

**Slettet:** s

**Slettet:** rating

**Slettet:** the

**Slettet:** t

**Slettet:** +

**Slettet:** u

**Slettet:** hazard

**Slettet:** (

**Slettet:** )

**Slettet:** *(1)*

**Slettet:** *!*

**Slettet:** a

**Slettet:** both expert and recreational

**Slettet:** p

5. *We need to teach people dynamics and terrain traps.* Problem: A relatively large share (a third) find it difficult to manage others in the group, many also find terrain traps problematic. Possible solution: Use the "avalanche school" to educate users about terrain traps and talk about group dynamics to help users make better choices about whom they choose to recreate with in avalanche terrain.

6. *Avalanche problem is important*. Problem: The avalanche danger level is not enough for making decisions in avalanche terrain, more detailed information is needed. Possible solution: Promote the avalanche problem, especially at danger levels 2 and 3, which also are the conditions most fatalities occur. Streamline the presentation of the avalanche problem according to danger level and reduce overlap with the avalanche danger assessment in order to reduce the complexity for users. Reduce the amount of information to users at higher danger levels. The danger level was rated as important, but somewhat difficult to understand. It is a simple numeric value, but is determined from relatively complex and subjective factors, and is probably difficult for users to understand and use.

7. *Keep the EAWS symbols.* The users considered the new EAWS standard icons for the avalanche problems to communicate the danger well, although the users were not familiar with the icons.

In conclusion, our study has confirmed that the communication of the avalanche danger on Varsom.no is perceived as effective by the users. The results of the testing of the effectiveness of different alternatives for communication of level 2 and level 4 avalanche danger suggested that the avalanche problems communicated more effectively than the danger level at lower danger levels. At the higher danger levels, no significant difference was found between the alternatives. The results suggest that a simple danger level is not enough to convey the intended warning message on lower danger levels, rather the warning should present the avalanche problem with a reasonable level of details. At higher danger levels, the results suggest that leaving out the advice and explanation resulted in a lower comprehension. For danger level 2 a user's competence mattered when it came to the rating of the alternatives, but not for danger level 4. Many users (67 %) and most NAWS experts (77 %) rated the avalanche problem as the most important element of the warning.

Based on the findings in this study, NAWS redesigned the avalanche warning on Varsom.no: the communication of the core sectors was improved (displayed in red signal colour rather that vague grey), a location search function was added, the display of avalanche problems was moved up to just below the main message, the redundancy in information between the avalanche problem, snow cover analysis and the avalanche danger assessment was reduced, the region map was relocated down to the bottom of the page and the mountain weather and snow cover analysis were restructured.

Norwegian users, experts and avalanche warnings were used in this study, but we believe the methods and results are important to the wider scientific community and AWS' in other countries. The building blocks and communication techniques of the avalanche warnings on Varsom.no follows the standards of EAWS, as Varsom.no and NAWS were developed in collaboration with a number of AWS' in Europe and North America.

Our study sheds light on how effectively key information is communicated in avalanche warnings. However, we recommend more studies on communication and impact of avalanche warnings, including in-the-field-testing, testing of the use of

**Slettet:** *(2)!*

**Slettet:** more than

**Slettet:** identify terrain traps, and to

**Slettet:**

**Slettet:** the

**Slettet:** under

**Slettet:** recreational

**Slettet:** rating

**Slettet:** Most experts (79 %) and m

**Slettet:** recreational

**Slettet:** 5

**Slettet:** 9

**Slettet:** and

**Slettet:** the

**Slettet:** and

**Slettet:** was

avalanche problems with regards to people and terrain choices, and to further develop methods for quantifying the effectiveness of such communication.

**Acknowledgement**

The three referees Benjamin Zweifel, Mark Staples and anonymous provided constructive and very valuable comments, which helped the authors to improve the manuscript. Also comments provided by Frank Techel were of great value when revising the manuscript. The editor Sven Fuchs ensured an expedient and fair revision process. Our colleagues Ingeborg Kleivane, Erik Johnsen, Jostein Aasen, Solveig Kosberg and Espen Nordahl provided valuable feedback during the study.
The study is the result of an initiative at the Center for Avalanche Research and Education, which is a collaboration between the UiT The Arctic University of Norway and the Norwegian Water Resources and Energy Directorate. The authors express their gratitude for the support provided by the chairman of the CARE steering committee, Dr. Carsten Roland.
The feedback from the users of Varsom and the NAWS experts made this study possible, thanks to all of you!

**References**

[revised manuscript text omitted]